# Activity Grammars for Temporal Action Segmentation

**Dayoung Gong**\*    **Joonseok Lee**\*    **Deunsol Jung**    **Suha Kwak**    **Minsu Cho**

Pohang University of Science and Technology (POSTECH)

{dayoung.gong, jameslee, deunsol.jung, suha.kwak, mscho}@postech.ac.kr

http://cvlab.postech.ac.kr/research/KARI

## Abstract

Sequence prediction on temporal data requires the ability to understand compositional structures of multi-level semantics beyond individual and contextual properties. The task of temporal action segmentation, which aims at translating an untrimmed activity video into a sequence of action segments, remains challenging for this reason. This paper addresses the problem by introducing an effective activity grammar to guide neural predictions for temporal action segmentation. We propose a novel grammar induction algorithm that extracts a powerful context-free grammar from action sequence data. We also develop an efficient generalized parser that transforms frame-level probability distributions into a reliable sequence of actions according to the induced grammar with recursive rules. Our approach can be combined with any neural network for temporal action segmentation to enhance the sequence prediction and discover its compositional structure. Experimental results demonstrate that our method significantly improves temporal action segmentation in terms of both performance and interpretability on two standard benchmarks, Breakfast and 50 Salads.

## 1 Introduction

Human activities in videos do not proceed by accident; they are structured being subject to generative rules imposed by the goal of activities, the properties of individual actions, the physical environment, and so on. Comprehending such a compositional structure of multi-granular semantics in human activity poses a significant challenge in video understanding research. The task of temporal action segmentation, which aims at translating an untrimmed activity video into a sequence of action segments, remains challenging due to the reason. The recent methods based on deep neural networks [25, 9, 45, 2, 15, 16, 1] have shown remarkable improvement in learning temporal relations of actions in an implicit manner, but often face out-of-context errors that reveal the lack of capacity to capture the intricate structures of human activity, and the scarcity of annotated data exacerbates the issue in training. In this work, we address the problem by introducing an effective activity grammar to guide neural predictions for temporal action segmentation.

Grammar is a natural and powerful way of explicitly representing the hierarchical structure of languages [14] and can also be applied to express the structure of activities. Despite the extensive body of grammar-based research for video understanding [23, 24, 33, 35, 32], none of these approaches have successfully integrated recursive rules. Recursive rules are indispensable for expressing complex and realistic structures found in action phrases and activities. To achieve this, we introduce a novel activity grammar induction algorithm, *Key-Action-based Recursive Induction* (KARI), that extracts a powerful probabilistic context-free grammar while capturing the characteristics of the activity. Since an activity is composed of multiple actions, each activity exhibits a distinctive temporal structure based on pivotal actions, setting it apart from other activities. The proposed grammar induction

---

\*Equal contribution.

37th Conference on Neural Information Processing Systems (NeurIPS 2023).

enables recursive rules with flexible temporal orders, which leads to powerful generalization capability. We also propose a novel activity grammar evaluation framework to evaluate the generalization and discrimination power of the proposed grammar induction algorithm. To incorporate the induced activity grammar into temporal action segmentation, we develop an effective parser, dubbed BEP, which searches the optimal rules according to the classification outputs generated by an off-the-shelf action segmentation model. Our approach can be combined with any neural network for temporal action segmentation to enhance the sequence prediction and discover its compositional structure.

The main contribution of this paper can be summarized as follows:

- We introduce a novel grammar induction algorithm that extracts a powerful context-free grammar with recursive rules based on key actions and temporal dependencies.
- We develop an effective parser that efficiently handles recursive rules of context-free grammar by using Breadth-first search and pruning.
- We propose a new grammar evaluation framework to assess the generalization and discrimination capabilities of the induced activity grammars.
- We show that the proposed method significantly improves the performance of temporal action segmentation models, as demonstrated through a comprehensive evaluation on two benchmarks, Breakfast and 50 Salads.

## 2 Related work

**Grammar for activity analysis.** Grammar is an essential tool to represent the compositional structure of language [14] and has been mainly studied in the context of natural language processing (NLP) [21, 22, 20, 37]. Grammar has been extensively studied in various research areas [28, 29, 32, 8, 43, 13, 42, 11, 27, 12]. Similarly, a grammatical framework can be used to express the structure of activities. Several work [23, 24, 33, 35] have defined context-free grammars based on possible temporal transitions between actions for action detection and recognition. Vo and Bovick [41] propose a stochastic grammar to model a hierarchical representation of activity based on AND-rules and OR-rules. Richard et al. [34] propose a context-free grammar defined on action sequences for weakly-supervised temporal action segmentation. Qi et al. [30, 32, 31] utilize a grammar induction algorithm named ADIOS [37] to induce grammar from action corpus. However, none of the proposed grammar for activity analysis includes recursive rules, which are a fundamental factor in expressing repetitions of actions or action phrases. In this paper, we propose a novel action grammar for temporal action segmentation based on key action and temporal dependency between actions considering recursive temporal structure.

**Temporal action segmentation (TAS).** Various methods have been proposed to address the task. Early work utilizes temporal sliding windows [36, 19] to detect action segments, and language-based methods [24, 23] has been proposed to utilize a temporal hierarchy of actions during segmentation. Recently, a deep-learning-based model named the temporal convolutional networks (TCN) has been proposed with an encoder-decoder architecture [25, 9]. Moreover, transformer-based models [45, 2] are recently introduced to leverage global temporal relations between actions based on self-attention and cross-attention mechanisms [40]. Other researches have been proposed to improve the accuracy of temporal action segmentation based on existing models [9, 45]. Huang *et al.* [15] introduce a network module named Graph-based Temporal Reasoning Module (GTRM) that is applied on top of baseline models to learn temporal relations of action segments. Ishikawa *et al.* [16] suggest an action segment refinement framework (ASRF) dividing a task into frame-wise action segmentation and boundary regression. They refine frame-level classification results with the predicted action boundaries. Gao *et al.* [10] propose a global-to-local search scheme to find appropriate receptive field combinations instead of heuristic respective fields. Ahn and Lee [1] recently propose a hierarchical action segmentation refiner (HASR), which refines segmentation results by applying multi-granular context information from videos. A fast approximate inference method named FIFA for temporal action segmentation and alignment instead of dynamic programming is proposed by Souri *et al.* [38]. Other researches [5, 6] reformulate TAS as a cross-domain problem with different domains of spatio-temporal variations, introducing self-supervised temporal domain adaptation. Xu *et al.* [44] proposes differentiable temporal logic (DTL), which is a model-agnostic framework to give temporal constraints to neural networks. In this paper, we propose a neuro-symbolic approach where the activity grammar induced by the proposed grammar induction algorithm guides a temporal action segmentation model to refine segmental errors through parsing.

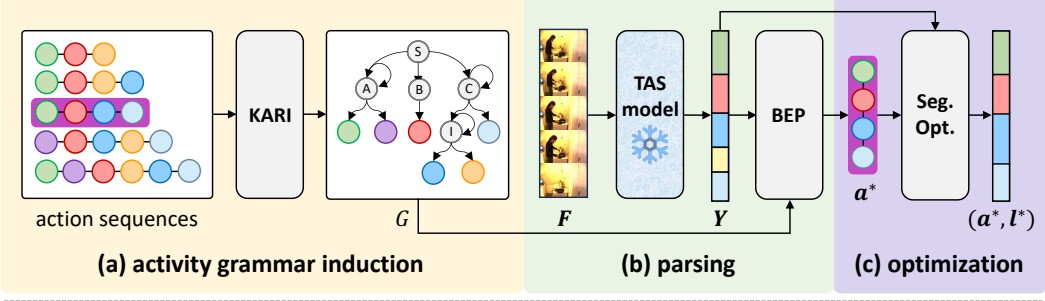

$G$: KARI-induced grammar, $F$: video, $Y$: prediction, $(a^*, l^*)$ : refined action (sequences, length), **Seg. Opt.**: segmentation optimization

Figure 1: **Overall pipeline of the proposed method.** (a) KARI induces an activity grammar $G$ from action sequences in the training data, (b) BEP parses neural predictions $Y$ from the off-the-shelf temporal action segmentation model given a video $F$ by using the KARI-induced grammar $G$, and (c) the final output of optimal action sequences and lengths $(a^*, l^*)$ is achieved through segmentation optimization. It is best viewed in color.

## 3 Our approach

Given a video of $T$ frames $F = [F_1, F_2, ..., F_T]$ and a predefined set of action classes $\mathcal{A}$, the goal of temporal action segmentation is to translate the video into a sequence of actions $a = [a_1, a_2, ..., a_N]$ and their associated frame lengths $l = [l_1, l_2, ..., l_N]$ where $N$ is unknown, $a_i \in \mathcal{A}$ for $1 \le i \le N$, $a_i \neq a_{i+1}$ for $1 \le i \le N-1$, and $\sum_{i=1}^{N} l_i = T$.[2] The resultant output of $a$ and $l$ indicates that the video consists of $N$ segments and each pair $(a_i, l_i)$ represents the action and length of $i_{\text{th}}$ segment.

In this work, we introduce an activity grammar that guides neural predictions for temporal action segmentation through parsing. We propose a novel activity grammar induction algorithm named KARI and an efficient parser called BEP. The overall pipeline of the proposed method consists of three steps, as illustrated in Fig. 1. First of all, KARI induces an activity grammar from action sequences in the training data. Using the KARI-induced grammar, BEP then takes the frame-level class prediction $Y \in \mathbb{R}^{T \times |\mathcal{A}|}$ from the off-the-shelf temporal action segmentation model [45, 9] and produces a grammar-consistent action sequence $a^*$. Finally, segmentation optimization is performed to obtain optimal action lengths $l^*$ based on $a^*$ and $Y$. In the following, we introduce the activity grammar as a probabilistic context-free grammar (Section 3.1), present KARI (Section 3.2) and BEP (Section 3.3), and describe a segmentation optimization method for final outputs (Section 3.4).

### 3.1 Activity grammar

We define the *activity grammar* as a probabilistic context-free grammar (PCFG) [17], designed to derive diverse action sequences pertaining to a specific activity class. The activity grammar, denoted as $G = (\mathcal{V}, \Sigma, \mathcal{P}, S)$, follows the conventional PCFG which consists of four components: a finite set of variables $\mathcal{V}$, a finite set of terminals $\Sigma$, a finite set of production rules $\mathcal{P}$, and the start symbol $S \in \mathcal{V}$. In our context, the set of terminals $\Sigma$ becomes the set of action classes $\mathcal{A}$, and the production rules $\mathcal{P}$ are used to generate action sequences from the start variable $S$. We use two types of production rules, 'AND' and 'OR', defined as follows:

$$\text{AND} : \quad V \to \alpha \qquad\qquad\qquad\qquad \text{where } V \in \mathcal{V} \text{ and } \alpha \in (\Sigma \cup \mathcal{V})^*, \qquad (1)$$
$$\text{OR} : \quad V \to V_1\,[p_1]\,|\,V_2\,[p_2]\,|\,\cdots\,|\,V_n\,[p_n] \quad \text{where } V, V_1, ..., V_n \in \mathcal{V}. \qquad (2)$$

The AND rule replaces a head variable $V$ with a sequence of variables and terminals $\alpha$, determining the order of the terminals and variables. In contrast, the OR rule converts a head variable $V$ to a sub-variable $V_i$ with the probability $p_i$, providing multiple alternatives for replacement; '|' denotes 'OR' operation. These two types of rules allow us to generate action sequences hierarchically.

### 3.2 Grammar induction: Key-Action-based Recursive Induction (KARI)

Grammar induction refers to the process of learning grammars from data [37]. In our context, it takes action sequences of a specific activity in the training set and produces an activity grammar that is

---

[2]In fact, this form of output is equivalent to that of frame-level action classification, which predict an action class for each frame, and the sequence of frame-level actions is easily converted to $(a, l)$ and vice versa.

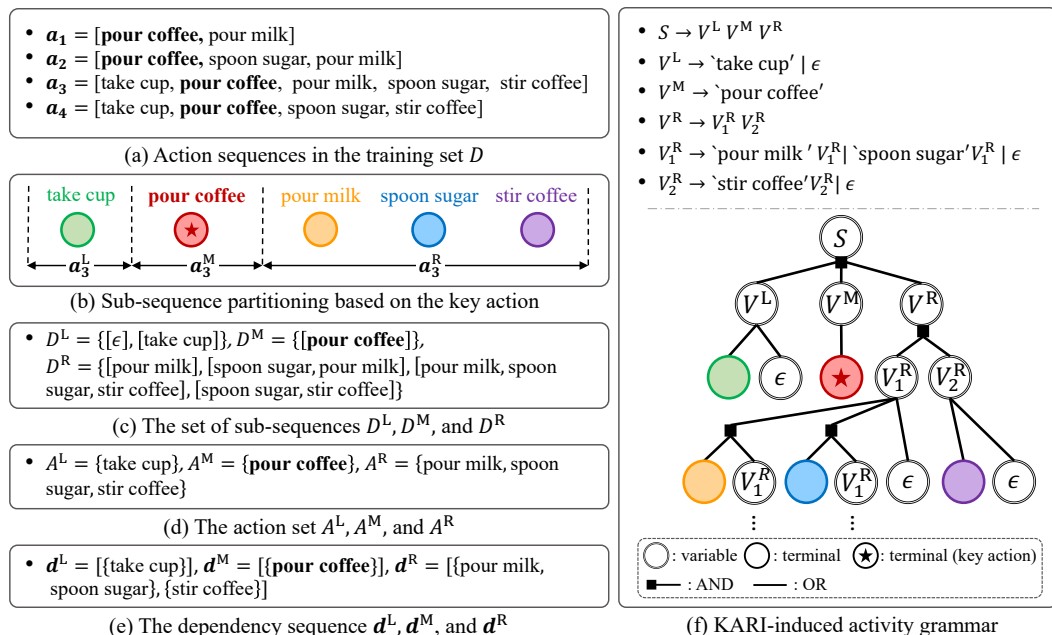

- $a_1$ = [**pour coffee**, pour milk]
- $a_2$ = [**pour coffee**, spoon sugar, pour milk]
- $a_3$ = [take cup, **pour coffee**, pour milk, spoon sugar, stir coffee]
- $a_4$ = [take cup, **pour coffee**, spoon sugar, stir coffee]

(a) Action sequences in the training set $D$

take cup    **pour coffee**    pour milk    spoon sugar    stir coffee

$\xleftarrow{\quad a_3^L \quad}$ $\xleftarrow{\quad a_3^M \quad}$ $\xleftarrow{\qquad\qquad a_3^R \qquad\qquad}$

(b) Sub-sequence partitioning based on the key action

- $D^L$ = {[$\epsilon$], [take cup]}, $D^M$ = {[**pour coffee**]}, $D^R$ = {[pour milk], [spoon sugar, pour milk], [pour milk, spoon sugar, stir coffee], [spoon sugar, stir coffee]}

(c) The set of sub-sequences $D^L$, $D^M$, and $D^R$

- $A^L$ = {take cup}, $A^M$ = {**pour coffee**}, $A^R$ = {pour milk, spoon sugar, stir coffee}

(d) The action set $A^L$, $A^M$, and $A^R$

- $d^L$ = [{take cup}], $d^M$ = [{**pour coffee**}], $d^R$ = [{pour milk, spoon sugar}, {stir coffee}]

(e) The dependency sequence $d^L$, $d^M$, and $d^R$

- $S \rightarrow V^L V^M V^R$
- $V^L \rightarrow$ `take cup' $| \epsilon$
- $V^M \rightarrow$ `pour coffee'
- $V^R \rightarrow V_1^R V_2^R$
- $V_1^R \rightarrow$ `pour milk ' $V_1^R|$ `spoon sugar' $V_1^R | \epsilon$
- $V_2^R \rightarrow$ `stir coffee' $V_2^R| \epsilon$

$\bigcirc$: variable   $\bigcirc$: terminal   $\bigstar$: terminal (key action)

$\blacksquare$ : AND    —— : OR

(f) KARI-induced activity grammar

Figure 2: **Example of activity grammar induction of KARI.** (a) Example action sequences are provided with 'pour coffee' as the key action with $N^{\text{key}}$ set to 1. (b) Action sequence, *e.g.*, $a_3$, is segmented into sub-sequences based on key actions. (c) All action sub-sequences $a^\Omega$ consist in a set of sub-sequences $\mathcal{D}^\Omega$. (d) The action set $\mathcal{A}^\Omega$ contains all the actions occurring in $\mathcal{D}^\Omega$. (e) Temporally independent actions are grouped, where each action group is temporally dependent in the action group sequence $d^\Omega$. (f) The resultant KARI-induced activity grammar is shown. For simplicity, we omit the probability, and it is best viewed in color.

able to parse action sequences of the activity; the induced grammar should be able to parse unseen sequences of the activity as well as the sequences in the training set for generalization. To obtain an effective activity grammar avoiding under-/over-generalization, we introduce two main concepts for grammar induction: *key action* and *temporal dependency*.

The key actions for a specific activity are those consistently present in every action sequence from the training dataset. Specifically, the top $N^{\text{key}}$ most frequently occurring actions among these are selected as the key actions. The hyperparameters of the number of key actions $N^{\text{key}}$ affects the degree of generalization achieved by the induced grammar. The temporal dependency refers to the relevance of temporal orders across actions. Temporally independent actions do not occur in a specific temporal order. This concept of temporal dependency can also be extended to groups of actions, meaning that some groups of actions can be temporally dependent on others.

We induce an activity grammar based on the key actions and the temporal dependency. Action sequences are divided into sub-sequences using the key actions as reference points, and the temporal dependencies between actions within the sub-sequences are established; temporally dependent actions are represented using AND rules (Eq. 1), while temporally independent actions are expressed with OR rules (Eq. 2). We give an example of grammar induction in Fig. 2; four action sequences are given in Fig. 2a, where the action class 'pour coffee' is chosen as the key action with the number of key actions $N^{\text{key}}$ set to 1.

Given the action sequences from the training dataset $\mathcal{D}$, we begin grammar induction by identifying a set of key actions $\mathcal{K} \subset \mathcal{A}$ with the pre-defined hyperparameter $N^{\text{key}}$. Using the key actions, each action sequence $a \in \mathcal{D}$ is divided into three parts: $a = [a^L, a^M, a^R]$. The sub-sequences $a^L$, $a^M$, and $a^R$ denote the portions of the original action sequence that occurred *before*, *between*, and *after* the key actions, respectively; the sub-sequence $a^M$ starts from the first key action and includes up to the last key action in $\mathcal{K}$. An example in Fig. 2b shows that the action sequence $a_3$ is divided into three sub-sequences using the key actions. For notational convenience, we will use the superscript $\Omega \in \{L, M, R\}$ to denote one of the three parts. All action sub-sequences $a^\Omega$ in a specific part $\Omega$ are grouped to consist in a corresponding set of sub-sequences $\mathcal{D}^\Omega$ (*cf.* Fig 2c). The action set $\mathcal{A}^\Omega \subseteq \mathcal{A}$

is then defined to contain all the actions occurring in $\mathcal{D}^\Omega$ (*cf.* Fig 2d). To determine the temporal dependencies among the actions of $\mathcal{A}^\Omega$, pairwise temporal orders are considered as follows. If one action always occurs before the other in $\mathcal{D}^\Omega$, then the two actions are temporally dependent and otherwise temporally independent. Based on the concepts, we construct the action group sequence $\boldsymbol{d}^\Omega$ by collecting the temporally independent actions as an action group and arranging such action groups according to their temporal dependencies (*cf.* Fig 2e).

In the following, we describe how to construct the production rules $\mathcal{P}$ of the activity grammar $G$.

**Start rule**. We first create the rule for the start variable $S$:

$$S \rightarrow V^{\mathrm{L}} V^{\mathrm{M}} V^{\mathrm{R}} , \tag{3}$$

where $V^{\mathrm{L}}$, $V^{\mathrm{M}}$, and $V^{\mathrm{R}}$ are variables used to derive left, middle, and right parts of the action sequence, respectively.

**Rule for the variable $V^\Omega$.** For $V^\Omega, \Omega \in \{\mathrm{L}, \mathrm{R}\}$, we construct an AND rule of action groups based on action group sequence $\boldsymbol{d}^\Omega$:

$$V^\Omega \rightarrow V_1^\Omega V_2^\Omega \cdots V_{|\boldsymbol{d}^\Omega|}^\Omega, \tag{4}$$

where the variable $V_i^\Omega$ represents the $i_{\mathrm{th}}$ action group in the action group sequence $\boldsymbol{d}_i^\Omega$. Since actions in an action group are considered temporally independent, we construct an OR rule for each action group:

$$V_i^\Omega \rightarrow d_{i,1}^\Omega V_i^\Omega \ [p_{i,1}^\Omega] \mid d_{i,2}^\Omega V_i^\Omega \ [p_{i,2}^\Omega] \mid \cdots \mid d_{i,|\boldsymbol{d}^\Omega|}^\Omega V_i \ [p_{i,|\boldsymbol{d}^\Omega|}^\Omega] \mid \epsilon \ [p_{i,\epsilon}^\Omega] , \tag{5}$$

where $d_{i,j}^\Omega$ denotes the $j_{\mathrm{th}}$ action from the action group $\boldsymbol{d}_i^\Omega$. The variable $V_i^\Omega$ yields $d_{i,j}^\Omega V_i^\Omega$ with the probability $p_{i,j}^\Omega$. This rule can be recursively used to proceed to the variable $V_i^\Omega$ in the next step. This recursive structure allows for repeated selection of actions within the same action group, leading to the generation of diverse action sequences, which is effective for generalization. To avoid an infinite loop of the recursion, the empty string $\epsilon$ with the escape probability $p_{i,\epsilon}^\Omega$ is added to Eq. 5. For the details, refer to the transition probability $p_{i,j}^\Omega$ and the escape probability $p_{i,\epsilon}^\Omega$ in Appendix A.1.

**Rule for the middle variable $V^{\mathrm{M}}$.** Since the temporal order of key actions might vary, we consider all the possible temporal orders between key actions in $\mathcal{K}$. A set of temporal permutations of actions is denoted as $\Pi$, where each possible temporal permutation is represented by the OR rule:

$$V^{\mathrm{M}} \rightarrow V_1^{\mathrm{M}} [p_1^{\mathrm{M}}] \mid V_2^{\mathrm{M}} [p_2^{\mathrm{M}}] \mid \cdots \mid V_{|\Pi|}^{\mathrm{M}} [p_\Pi^{\mathrm{M}}] \mid \epsilon [p_\epsilon^{\mathrm{M}}]. \tag{6}$$

The rule for the permutation variable $V_i^{\mathrm{M}}$ is defined by the AND rule:

$$V_i^{\mathrm{M}} \rightarrow \pi_{i,1} V^{\mathrm{M}(i,1)} \cdots \pi_{i,|\boldsymbol{\pi}_i|} V^{\mathrm{M}(i,|\boldsymbol{\pi}_i|)} V^{\mathrm{M}} , \tag{7}$$

where all the key actions are included. Note that $\pi_{i,j}$ represents the $j_{\mathrm{th}}$ action of the permutation $\boldsymbol{\pi}_i \in \Pi$, and the variable $V^{\mathrm{M}(i,j)}$ derives action sub-sequences between actions $\pi_{i,j}$ and $\pi_{i,j+1}$. The production rule for $V^{\mathrm{M}(i,j)}$ adheres to the rules specified in Eq. 4 and 5. The resultant KARI-induced grammar from the example is shown in Fig. 2f, highlighting the compositional structure of actions.

### 3.3  Parser: Breadth-first Earley Parser (BEP)

The goal of the parser is to identify the optimal action sequence $\boldsymbol{a}^*$ by discovering the most likely grammatical structure based on the output of the action segmentation model [9, 45]. In other words, the parser examines the production rules of the activity grammar to determine whether the given neural prediction $\boldsymbol{Y}$ can be parsed by the grammar $G$. However, when the grammar includes recursive rules, the existing parser struggles to complete the parsing within a reasonable time due to the significant increase in branches from the parse tree. To address this challenge, we introduce an effective parser dubbed BEP, integrating Breadth-first search (BFS) and a pruning technique into a generalized Earley parser (GEP) [32]. Since the BFS prioritizes production rules closer to the start variable, it helps the parser understand the entire context of the activity before branching to recursive iterations. Simultaneously, pruning effectively reduces the vast search space generated by OR nodes and recursion, enabling the parser to focus on more relevant rules for the activity.

For parsing, we employ two heuristic probabilities introduced in [32] to compute the probability of variables and terminals within the parse tree. Specifically, let $\boldsymbol{Y}_{t,x}$ denote the probability of frame $t$

being labeled as $x$. In this context, we denote the last action in the action sequence $\boldsymbol{a}$ as $x$, *i.e.* $x = a_N$, where $\boldsymbol{a} = [a_1, a_2, ..., a_N]$, for simplicity. The transition probability $g(x \mid \boldsymbol{a}_{1:N-1}, G)$ determines the probability of parsing action $x$ given the $\boldsymbol{a}_{1:N-1}$ and the grammar $G$.

The parsing probability $p(\boldsymbol{F}_{1:T} \to \boldsymbol{a} \mid G)$ computes the probability of $\boldsymbol{a}$ being the action sequence for $\boldsymbol{F}_{1:T}$. The probability at $t = 1$ is initialized by:

$$p(F_1 \to \boldsymbol{a} \mid G) = \begin{cases} g(x \mid \epsilon, G)\, \boldsymbol{Y}_{1,x} & \text{if } \boldsymbol{a} \text{ contains only } x, \\ 0 & \text{otherwise,} \end{cases} \tag{8}$$

where $\epsilon$ indicates an empty string.

Since we assume that the last action of $\boldsymbol{a}$ is classified as $x$, the parsing probability $p(\boldsymbol{F}_{1:t} \to \boldsymbol{a} \mid G)$ can be represented with the probability of the previous frames:

$$p(\boldsymbol{F}_{1:t} \to \boldsymbol{a} \mid G) = \boldsymbol{Y}_{t,x}(\, p(\boldsymbol{F}_{1:t-1} \to \boldsymbol{a} \mid G) + g(x \mid \boldsymbol{a}_{1:N-1}, G)\, p(\boldsymbol{F}_{1:t-1} \to \boldsymbol{a}_{1:N-1} \mid G)\,). \tag{9}$$

The prefix probability $p(\boldsymbol{F}_{1:T} \to \boldsymbol{a}... \mid G)$ represents the probability of $\boldsymbol{a}$ being the prefix of $\boldsymbol{a}^*$. This probability is computed by measuring the probability that $\boldsymbol{a}$ is the action sequence for the frame $\boldsymbol{F}_{1:t}$ with $t$ in the range $[1, T]$:

$$p(\boldsymbol{F}_{1:T} \to \boldsymbol{a}... \mid G) = p(F_1 \to \boldsymbol{a} \mid G) + g(x \mid \boldsymbol{a}_{1:N-1}, G) \sum_{t=2}^{T} \boldsymbol{Y}_{t,x}\, p(\boldsymbol{F}_{1:t-1} \to \boldsymbol{a}_{1:N-1} \mid G). \tag{10}$$

The parsing operation is structured following the original Earley parser [7], consisting of three key operations: prediction, scanning, and completion. These operations involve the update and generation of states, where every state comprises the rule being processed, the parent state, the parsed action sequence denoted as $\boldsymbol{a}$, and the prefix probability denoted as $p(\boldsymbol{a}...)$. The states are enqueued and prioritized by their depth $d$ within the parse tree.

- **Prediction**: for every state $Q(m, n, d)$ of the form $(A \to \alpha \cdot B\beta, Q(i, j, k), \boldsymbol{a}, p(\boldsymbol{a}...))$, add $(B \to \cdot\Gamma, Q(m, n, d), \boldsymbol{a}, p(\boldsymbol{a}...))$ to $Q(m, n, d+1)$ for every production rule in the grammar with $B$ on the left-hand side.
- **Scanning**: for every state in $Q(m, n, d)$ of the form $(A \to \alpha \cdot w\beta, Q(i, j, k), \boldsymbol{a}, p(\boldsymbol{a}...))$, append the new terminal $w$ to $\boldsymbol{a}$ and compute the probability $p((\boldsymbol{a} + w)...)$. Create a new set $Q(m+1, n', d)$ where $n'$ is the current size of $Q(m+1)$. Add $(A \to \alpha w \cdot \beta, Q(i, j, k), \boldsymbol{a} + w, p((\boldsymbol{a} + w)...))$ to $Q(m+1, n', d)$.
- **Completion**: for every state in $Q(m, n, d)$ of the form $(A \to \Gamma\cdot, Q(i, j, k), \boldsymbol{a}, p(\boldsymbol{a}...))$, find states in $Q(i, j, k)$ of the form $(B \to \alpha \cdot A\beta, Q(i', j', k'), \boldsymbol{a}', p(\boldsymbol{a}'...))$ and add $(B \to \alpha A \cdot \beta, Q(i', j', k'), \boldsymbol{a}, p(\boldsymbol{a}...))$ to $Q(m, n, d-1)$.

The symbols $\alpha$, $\beta$, and $\Gamma$ represent arbitrary strings consisting of terminals and variables, *i.e.* $\alpha, \beta, \Gamma \in (\Sigma \cup V)^*$. The symbols $A$ and $B$ refer to the variables, while $w$ denotes a single terminal. The symbol $Q$ represents the set of states, and the dot $(\cdot)$ denotes the current position of the parser within the production rule.

Additionally, we introduce a pruning technique of limiting the queue size to reduce the vast search space in the parse tree, similar to the beam search. Specifically, the parser preserves only the top $N^{\text{queue}}$ elements from the queue in order of the parsing probability of each state. The parsing process terminates when the parser identifies that the parsed action sequence $\boldsymbol{a}^*$ has a higher parsing probability than the prefix probabilities of any other states in the queue. For the further details, refer to Appendix B.

## 3.4 Segmentation optimization

The objective of segmentation optimization is to determine the optimal alignment between the input classification probability matrix $\boldsymbol{Y}$ and the action sequence $\boldsymbol{a}^*$. In other words, the entire frames are allocated within the action sequences $\boldsymbol{a}^* = [a_1^*, a_2^*, ..., a_N^*]$, obtained from the parser, to determine the optimal action lengths $\boldsymbol{l}^* = [l_1^*, l_2^*, ..., l_N^*]$. In this work, we utilize dynamic programming-based Viterbi-like algorithm [35] for activity parsing. Similar to [26, 32], the optimizer explores all possible allocations and selects the one with the maximum product of probabilities:

$$\boldsymbol{l}^* = \arg\max_{\boldsymbol{l}}(p(\boldsymbol{l} \mid \boldsymbol{a}^*, \boldsymbol{Y}_{1:T})), \tag{11}$$

$$p(\boldsymbol{l} \mid \boldsymbol{a}, \boldsymbol{Y}_{1:t}) = \max_{i<t}(p(\boldsymbol{l}_{1:N-1} \mid \boldsymbol{a}_{1:N-1}, \boldsymbol{Y}_{1:i}) \prod_{j=i}^{t} \boldsymbol{Y}_{j,a_N}). \tag{12}$$

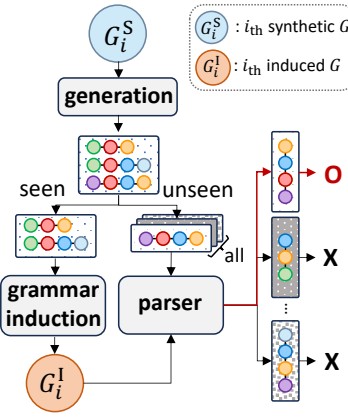

Figure 3: **Grammar evaluation**



Figure 4: **Confusion matrix of activity grammars.** The results of KARI-induced grammar are similar to the synthetic grammar, showing high recall with comparable precision.

Table 1: **Synthetic $G$ I**

| grammar | precision | recall |
|---|---|---|
| ADIOS-AND | 0.97 | 0.08 |
| ADIOS-OR | **0.99** | 0.25 |
| KARI (ours) | 0.93 | **0.98** |

Table 2: **Synthetic $G$ II**

| grammar | precision | recall |
|---|---|---|
| ADIOS-AND | 0.98 | 0.06 |
| ADIOS-OR | **1.00** | 0.33 |
| KARI (ours) | 0.92 | **0.96** |

## 4 Experimental evaluation and analysis

### 4.1 Datasets and evaluation metrics

**Datasets.** We conduct experiments on two widely used benchmark datasets for temporal action segmentation: Breakfast [23] and 50 Salads [39]. The Breakfast dataset, consisting of 1,712 videos, involves 52 individuals preparing 10 different breakfast activities comprised of 48 actions in 18 different kitchens. Similarly, the 50 Salads dataset comprises 50 egocentric videos of people preparing salads of a single activity with 17 fine-grained actions from 25 people. We used I3D [4] features provided by [9].

**Evaluation metrics.** For evaluation metrics, we report edit score, F1@$\{10, 25, 50\}$ scores, and frame-wise accuracy following the previous work [9, 45].

### 4.2 Implementation details

For KARI, we set the hyperparameters of the number of key actions $N^{\mathrm{key}}$ to 4 for Breakfast, and 3 for 50 Salads. We individually induce separate activity grammar for the ten activity classes within Breakfast and subsequently merge them into a unified grammar. For the comparison with the existing grammar used in the previous work [32, 31], we induce activity grammars of ADIOS [37] provided by [31]. Two types of ADIOS-induced grammar are induced: ADIOS-AND-induced grammar, primarily composed of AND rules with limited generalization capabilities, and ADIOS-OR-induced grammar, predominantly incorporating OR rules, offering improved generalization. Please refer to Appendix C.1 for grammar induction details.

For BEP, we configured the queue size $N^{\mathrm{queue}}$ to be 20. For efficiency, we adjust the sampling rate of the input video features to 50 for Breakfast and 100 for 50 Salads. We use two widely used models for the temporal action segmentation: ASFormer [45] based on Transformer and MS-TCN [9] based on CNNs. Since we apply the proposed method to the reproduced temporal action segmentation models, we directly compare and evaluate the performance based on the reproduced results.

### 4.3 Evaluation framework for activity grammar

We propose a novel evaluation framework to assess the generalization and discrimination capabilities of the activity grammar. Figure 3 shows the overall process of the grammar evaluation framework. We first generate a set of synthetic activity grammars $\mathcal{G}^{\mathrm{S}}$ randomly. Action sequences $\boldsymbol{a} \in \mathcal{D}_i^{\mathrm{all}}$ are generated from each synthetic grammar $G_i^{\mathrm{S}} \in \mathcal{G}^{\mathrm{S}}$, and these sequences are randomly divided into two sets: seen *seen* and *unseen*. For each seen set, a grammar induction algorithm is applied, resulting in the induced grammar $G_i^{\mathrm{I}}$ consisting in a corresponding set of induced grammars $\mathcal{G}^{\mathrm{I}}$. For grammar evaluation, the induced grammar $G_i^{\mathrm{I}} \in \mathcal{G}^{\mathrm{I}}$ parses action sequences from the entire unseen sets. The induced grammar should accurately parse the action sequences generated by the original synthetic grammar from which it was induced, while also effectively discriminating those generated by other synthetic grammars.

To simulate real-world video action sequences, we generate the synthetic activity grammars assuming temporal dependencies across actions. This indicates that certain actions follow a temporal order

Table 3: **The performance comparison on 50Salads**

| model | re-prod. | refine-ment | grammar induction | edit | F1@10 | F1@25 | F1@50 | acc. |
|---|---|---|---|---|---|---|---|---|
| ASFormer [45] | - | - | - | 75.0 | 76.0 | 70.6 | 57.4 | 73.5 |
| | ✓ | - | - | 76.5 | 83.8 | 81.7 | 74.8 | **86.1** |
| | ✓ | ✓ | ADIOS-AND | 58.3 | 70.0 | 68.0 | 59.4 | 76.2 |
| | ✓ | ✓ | ADIOS-OR | 61.1 | 72.0 | 70.1 | 62.4 | 78.9 |
| | ✓ | ✓ | KARI | **79.9** | **85.4** | **83.8** | **77.4** | 85.3 |
| MS-TCN [9] | - | - | - | 67.9 | 76.3 | 74.0 | 64.5 | 80.7 |
| | ✓ | - | - | 62.4 | 69.5 | 65.3 | 55.7 | 75.2 |
| | ✓ | ✓ | ADIOS-AND | 56.8 | 66.4 | 63.8 | 52.9 | 72.5 |
| | ✓ | ✓ | ADIOS-OR | 61.9 | 69.1 | 66.9 | 57.2 | 74.2 |
| | ✓ | ✓ | KARI | **66.7** | **75.1** | **73.2** | **60.8** | **76.7** |

Table 4: **The performance comparison on Breakfast**

| model | re-prod. | refine-ment | grammar induction | edit | F1@10 | F1@25 | F1@50 | acc. |
|---|---|---|---|---|---|---|---|---|
| ASFormer [45] | - | - | - | 75.0 | 76.0 | 70.6 | 57.4 | 73.5 |
| | ✓ | - | - | 75.6 | 77.3 | 72.0 | 59.4 | **74.3** |
| | ✓ | ✓ | ADIOS-AND | 69.2 | 69.8 | 64.9 | 52.2 | 72.4 |
| | ✓ | ✓ | ADIOS-OR | 70.3 | 71.8 | 66.8 | 54.2 | 71.8 |
| | ✓ | ✓ | KARI | **77.8** | **78.8** | **73.7** | **60.8** | 74.0 |
| MS-TCN [9] | - | - | - | 61.7 | 52.6 | 48.1 | 37.9 | 66.3 |
| | ✓ | - | - | 69.7 | 70.7 | 65.1 | 52.6 | **69.4** |
| | ✓ | ✓ | ADIOS-AND | 68.0 | 66.7 | 61.0 | 48.0 | 68.4 |
| | ✓ | ✓ | ADIOS-OR | 69.6 | 69.2 | 63.3 | 50.3 | 68.2 |
| | ✓ | ✓ | KARI | **74.9** | **74.6** | **68.7** | **55.1** | 68.8 |

while others do not adhere to such dependencies. To prevent parsing failures arising from uncovered terminals, we maintain a consistent set of terminals throughout the entire grammar while randomly assigning key actions to these terminals. The number of variables is randomly determined for each synthetic grammar. As evaluation metrics, we use *precision* and *recall* similar to the previous work [37, 3]. For the induced grammar $G_i^{\mathrm{I}}$, action sequences successfully parsed from the synthetic grammar $G_i^{\mathrm{S}}$ are classified as positive samples from the entire unseen sets, otherwise considered negative samples.

**Details.** In our experiment, we generate a total of 100 grammars, each consisting of 20 variables and 20 terminals. We have developed two types of synthetic grammars that differ in terms of temporal hierarchical difficulty. In synthetic grammar I, each terminal is allocated to a single variable, while in synthetic grammar II, terminals are randomly assigned multiple times to different variables. Three types of grammars are evaluated: induced by ADIOS-AND, ADIOS-OR, and proposed KARI.

**Results.** Table 1 and Table 2 show the results of grammar evaluation by using synthetic grammar I and II, respectively. Our activity grammar demonstrates robust generalization performances, achieving a recall of approximately 1.0 on unseen action sequences compared to others, maintaining comparable precision. The ADIOS-OR-induced grammar shows better generalization ability compared to the ADIOS-AND-induced grammar. We visualize a confusion matrix of the three types of grammar: synthetic grammar, KARI-induced grammar, and ADIOS-OR-induced grammar, as shown in Fig. 4. The confusion matrix shows the parsing accuracy of each unseen set over the synthetic grammar II. Higher accuracy is represented by brighter cells in the matrix. KARI-induced grammars demonstrate similar patterns in their confusion matrix compared to the synthetic grammars. This similarity indicates their capacity to generalize to unseen sets from which each grammar is induced, allowing effective discrimination of action sequences from other synthetic grammars.

## 4.4 Effects of the grammar-based refinement on temporal action segmentation

Table 3 and Table 4 show the performance of applying the proposed method to temporal action segmentation models [9, 45] across two benchmark datasets. The first row in Table 3 and Table 4 indicates the performance from the original paper [9, 45], whereas the second row represents the

Table 5: **Ablation study of KARI on 50 Salads**. Using both key action and recursive rules is effective for refining neural predictions from the temporal action segmentation models.

| refinement | key actions | recursive rules | edit | F1@10 | F1@25 | F1@50 | acc. |
|---|---|---|---|---|---|---|---|
| ✓ | ✓ | ✓ | **79.9** | **85.4** | **83.8** | **77.4** | **85.3** |
| ✓ | ✓ | - | 69.2 (10.7↓) | 77.1 (8.3↓) | 74.9 (8.9↓) | 67.4 (10.0↓) | 80.9 (4.4↓↓) |
| ✓ | - | - | 62.6 (17.3↓) | 72.9 (12.5↓) | 70.5 (13.3↓) | 63.0 (14.1↓) | 78.8 (6.5↓) |

Table 6: **BEP vs. GEP.** BEP is effective under a fair comparison to GEP.

| parser | $N^{\text{queue}}$ | edit | F1@10 | F1@25 | F1@50 | acc. |
|---|---|---|---|---|---|---|
| GEP | 10 | 73.3 | 81.1 | 79.1 | 72.9 | 84.0 |
| | 20 | 72.1 | 80.5 | 79.1 | 72.3 | 83.9 |
| | 30 | 72.3 | 79.8 | 78.1 | 71.4 | 84.2 |
| BEP | 10 | 78.3 | 84.9 | 83.2 | 76.9 | 84.9 |
| | 20 | **79.9** | **85.4** | **83.8** | **77.4** | **85.3** |
| | 30 | 78.9 | 85.5 | 83.8 | 77.3 | 85.1 |

Table 7: **Ablation on** $N^{\text{key}}$. Using proper number of key actions matters.

| $N^{\text{key}}$ | edit | F1@10 | F1@25 | F1@50 | acc. |
|---|---|---|---|---|---|
| 1 | 73.0 | 81.5 | 79.9 | 72.6 | 83.5 |
| 2 | 77.8 | 85.1 | 83.5 | 77.1 | **85.8** |
| 3 | **79.9** | **85.4** | **83.8** | **77.4** | 85.3 |
| 4 | 74.3 | 82.2 | 80.5 | 73.5 | 83.3 |
| 5 | 71.2 | 78.6 | 76.3 | 68.6 | 80.4 |
| 6 | 68.4 | 75.2 | 73.3 | 64.2 | 77.9 |

reproduced performance obtained using official codes. The comparison between the second and the last row of each compartment in each table reveals significant improvements in both edit scores and F1 scores. This result validates the effectiveness of leveraging activity grammars to refine segment-wise classification. Remarkably, the KARI-induced grammar shows great performance compared to both ADIOS-induced grammars, demonstrating the importance of generalizing the grammar to cover unseen action sequences during inference effectively.

## 4.5 Analysis

**Ablation studies of KARI.** Ablation studies of KARI are conducted on the 50 Salads dataset using ASFormer [45], as shown in Table 5 to demonstrate the effectiveness of each component, including *key actions* and *temporal dependency*. The results show that both key actions and recursive rules contribute to the significant improvement of grammar-based refinement. In particular, using recursive rules is essential for the activity grammar to be generalized to the unseen action sequences.

**BEP vs. GEP.** Table 6 presents the performance comparison of GEP and BEP using KARI-induced grammar. We limit the queue size of both parsers, as the parser without the limitation fails to complete parsing within a reasonable time. The results indicate that our BEP outperforms GEP under the same condition. This is attributed to GEP prioritizing states based on the highest probability, which increases the risk of getting trapped in local optima when performing selective pruning within specific branches. In contrast, BEP, which prioritizes low-depth states, allows for easier escape from cycles and OR nodes, contributing to improved overall performance.

**The number of key actions.** Table 7 shows the results by adjusting the number of key actions $N^{\text{key}}$ of KARI on 50 Salads. We set the value of $N^{\text{key}}$ ranging from 1 to 6, where the induced grammar with a smaller value generates the larger activity corpus. We find that setting $N^{\text{key}}$ to 3 outperforms the others, demonstrating the importance of achieving an appropriate level of generalization for effective refinement. Both excessive and insufficient generalization can negatively impact performance, highlighting the need to strike a balance in the generalization ability of the activity grammar.

**Grammar evaluation on real data.** We evaluate the parsing recall on the unseen action sequences of the Breakfast dataset. The results present the average recall across all splits for each activity. The number inside brackets indicates the average length of action sequences of each activity in $\mathcal{D}$. Table 8 compares the generalization capability of the five grammar induction algorithms [23, 35, 37], including KARI (details in Appendix C.1). The result demonstrates that KARI-induced grammar shows better generalization ability on real data compared to others. Remarkably, the KARI-induced grammar shows robust performance with the extended average length of the action sequences, whereas other algorithms exhibit poor generalization.

## 4.6 Qualitative results

Figure 5 presents a visual representation of the refined segmentation results on benchmark datasets. The proposed method successfully parses and identifies the actions 'pour oil' (red bar in Fig. 5a)

Table 8: **Grammar evaluation on real data.** We evaluate the proposed KARI-induced-grammar on Breakfast, demonstrating the superior high recall on unseen action sequences from each activity. The average length of action sequences of each activity is shown in parentheses.

| Grammar induction | scrambled egg (11.9) | pancake (11.1) | salad (9.9) | fried egg (9.5) | juice (7.2) | coffee (6.7) | sandwich (6.0) | cereal (5.1) | milk (5.0) | tea (5.0) | total (7.7) |
|---|---|---|---|---|---|---|---|---|---|---|---|
| Kuehne *et al.* [23] | 0.25 | 0.24 | 0.0 | 0.32 | 0.53 | 0.80 | 0.63 | 0.96 | 0.78 | 0.91 | 0.53 |
| Richard *et al.* [35] | 0.25 | 0.24 | 0.0 | 0.32 | 0.53 | 0.80 | 0.63 | 0.96 | 0.78 | 0.91 | 0.54 |
| ADIOS-AND [37] | 0.25 | 0.24 | 0.0 | 0.32 | 0.53 | 0.80 | 0.63 | 0.96 | 0.78 | 0.91 | 0.54 |
| ADIOS-OR [37] | 0.39 | 0.30 | 0.37 | 0.53 | 0.55 | 0.80 | 0.73 | 0.96 | 0.78 | 0.92 | 0.63 |
| KARI | **0.84** | **0.71** | **0.90** | **0.70** | **0.77** | **1.00** | **0.91** | **0.96** | **0.90** | **0.98** | **0.87** |

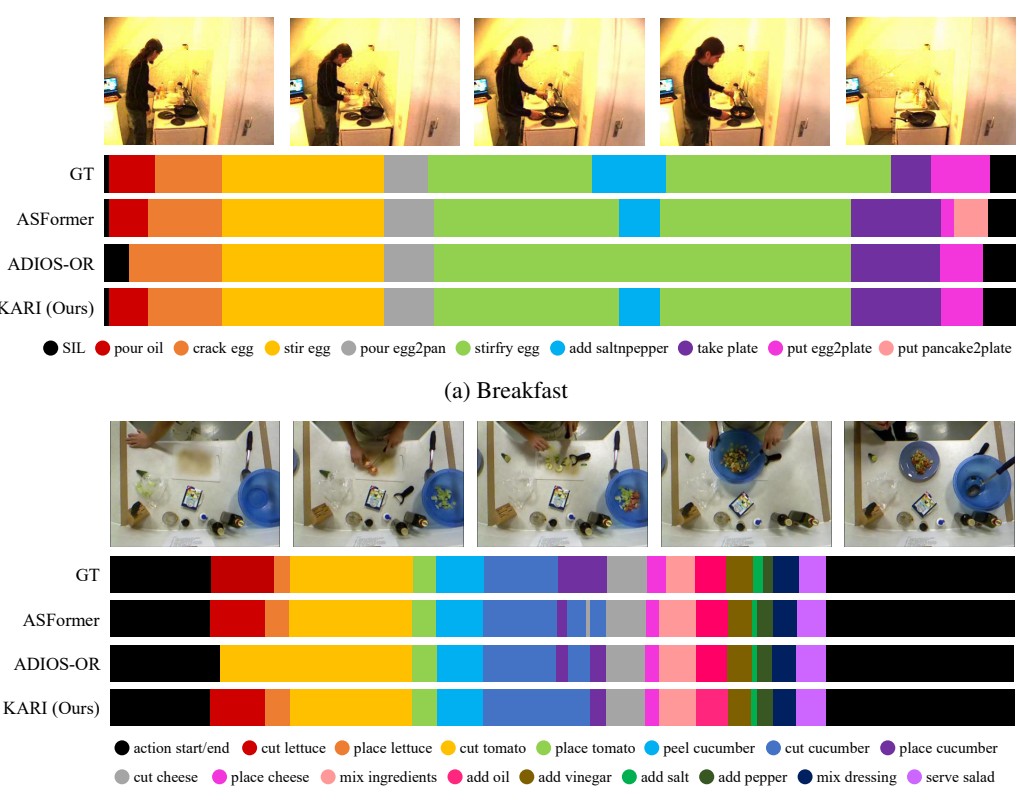

Figure 5: **Qualitative results.** KARI-induced grammar efficiently insert missing actions and removes out-of-context actions in ASFormer [45].

and 'add saltnpepper', (blue bar in Fig. 5a), which are omitted in the results obtained by using the ADIOS-OR induced grammar. The results show that KARI-induced grammar allows a more flexible temporal structure between actions. Furthermore, our method effectively removes actions such as 'put pancake2plate' that do not correspond to the intended activity. Similarly, qualitative results on 50 Salads in Fig. 5b show the effectiveness of the proposed method with complex action sequences. The overall results show that activity grammar-based refinement for the temporal action segmentation model is effective for correcting the neural predictions by using the grammar as a guide.

## 5 Conclusion

We have shown that the proposed approach enhances the sequence prediction and discovers its compositional structure, significantly improving temporal action segmentation in terms of both performance and interpretability. However, the improvement is limited by the initial output of the action segmentation network, which remains further research in the future. We believe that the grammar induction and parsing methods can be easily applied to other sequence prediction tasks.

# 6 Acknowledgements

This work was supported by the IITP grants (2022-0-00264: Comprehensive video understanding and generation with knowledge-based deep logic (50%), 2022-0-00290: Visual intelligence for space-time understanding and generation based on multi-layered visual common sense (20%), 2022-0-00959: Few-shot learning of causal inference in vision and language (20%), and 2019-0-01906: AI graduate school program at POSTECH (10%)) funded by the Korea government (MSIT).

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

# Appendices

In this supplement, we provide detailed descriptions of the proposed method and additional results, which are omitted in the main paper due to the lack of space. In Section A, we will describe the formulation of the probability of activity grammar. Algorithmic details of BEP is included in Section B. Section C compares KARI with the existing grammar induction algorithms for activity grammar and Section D presents additional qualitative results. We conclude this Appendix by discussing the broader impact of our research in Section E.

## A   Formulation of the probabilities in KARI

In this section, we describe the formulation of transition probability $p_{i,j}^{\Omega}$ and the escape probability $p_{i,\epsilon}^{\Omega}$ and $p_{\epsilon}^{M}$ in Eq. 5 and 6 in Section A.1. In the following, the derivation of the expectation of the escape probability is described in Section A.2.

### A.1   Formulation of the escape and transition probability

We first introduce and escape probabilities and the transition probabilities introduced in Eq. 5.
**Pre-processing.** Let $\boldsymbol{h}_i^{\Omega}$ represent a list of action sub-sequences, where the sub-sequence from $\boldsymbol{h}_i^{\Omega}$ removes actions that does not exist in the action group $\boldsymbol{d}_i^{\Omega}$ from the action sub-sequence in $\mathcal{D}^{\Omega}$. The empty string $\epsilon$ remains when the action sub-sequence does not include actions within the action group $\boldsymbol{d}_i^{\Omega}$. For example in Fig. 2, a list of sub-sequences $\boldsymbol{h}_1^{R}$ can be structured as $\boldsymbol{h}_1^{R} = [[\text{pour milk}], [\text{spoon sugar, pour milk}], [\text{pour milk, spoon sugar}], [\text{spoon sugar}]]$ with the corresponding action group $\boldsymbol{d}_1^{R} = \{\text{pour milk, spoon sugar}\}$. Similary, a list of sub-sequences $\boldsymbol{h}_2^{R}$ is structured as $\boldsymbol{h}_2^{R} = [\epsilon, \epsilon, [\text{stir coffee}], [\text{stir coffee}]]$ with the action group $\boldsymbol{d}_2^{R} = \{\text{stir coffee}\}$. This pre-processing step of generating $\boldsymbol{h}_i^{\Omega}$ enables us to consider the statistical probabilities associated with actions.

**Formulation of the escape probability.** The escape probability $p_{i,\epsilon}^{\Omega}$ and the transition probability $p_{i,j}^{\Omega}$ are both defined based on the number of recursion $n^{\text{rec}}$ of the current timestep; thereby these probabilities are represented as functions of $n^{\text{rec}}$. We first define the escape probability function:

$$p_{i,\epsilon}^{\Omega}(n^{\text{rec}}) = \begin{cases} \dfrac{\left|\left[\boldsymbol{a} \in \boldsymbol{h}_i^{\Omega} \mid \boldsymbol{a} = \epsilon\right]\right|}{|\boldsymbol{h}_i^{\Omega}|} & \text{if } n^{\text{rec}} = 1\,, \\ \dfrac{1}{\bar{N}^{\boldsymbol{h}_i^{\Omega}}} & \text{otherwise}\,, \end{cases} \tag{13}$$

where $\bar{N}^{\boldsymbol{h}_i^{\Omega}}$ is the average length of the sub-sequences in $\boldsymbol{h}_i^{\Omega}$. In the first recursion, *i.e.*, $n^{\text{rec}} = 1$, the probability calculation solely considers statistics of the actions. Otherwise, the probability is calculated based on the expected number of recursions, which will be introduced in Appendix A.2. In Fig. 2, $p_{1,\epsilon}^{R}(1) = 0$, since none of the action sub-sequence $\boldsymbol{a}$ from $\boldsymbol{h}_1^{R}$ is equal to the empty sequence, and $p_{1,\epsilon}^{R}(n^{\text{rec}} > 1) = \frac{2}{3}$, since the average length of sub-strings in $\boldsymbol{h}_1^{R}$ is 1.5.

**Formulation of the transition probability.** The action sequence $\boldsymbol{a} = [a_1, a_2, ..., a_N]$ represents the distinct action labels for the video segments, where $a_i \neq a_{i+1}$ as described in Section 3. In order to prevent the repetition of the same action in Eq. 5, we introduce an additional input $q$ when defining the transition probability. Here, $q$ refers to the index of the actions selected by the rule in the previous step, specifically at $(n^{\text{rec}} - 1)_{\text{th}}$ step where $n^{\text{rec}} > 1$. We simply put $q$ to 0 in the first recursion, *i.e.* $n^{\text{rec}} = 1$, which does not affect the results. The transition probability $p_{i,j}^{\Omega}$ is defined by:

$$p_{i,j}^{\Omega}(n^{\text{rec}}, q) = \begin{cases} \dfrac{\left|\left[\boldsymbol{a} \in \boldsymbol{h}_i^{\Omega} \mid a_1 = d_{i,j}^{\Omega}\right]\right|}{|\boldsymbol{h}_i^{\Omega}|} & \text{if } n^{\text{rec}} = 1\,, \\ 0 & \text{if } n^{\text{rec}} > 1 \text{ and } j = q, \\ \dfrac{p_{i,j}^{\Omega}(1,0)\left(1 - p_{i,\epsilon}^{\Omega}(n^{\text{rec}})\right)}{\sum_{l \neq q} p_{i,l}^{\Omega}(1,0)} & \text{otherwise.} \end{cases} \tag{14}$$

The escape probability $p_{\epsilon}^{M}$ and the transition probability $p_{i,j}^{M}$ for the middle variable $V^{M}$ in Eq. 6 is defined in the same way as Eq. 13 and Eq. 14, respectively.

## A.2 Derivation of the escape probability

We introduce the formulation of the escape probability $p_{i,\epsilon}^{\Omega}$ in Appendix A.1. The escape probability is required to avoid an infinite loop of the rules and guarantee the length of sequences from the recursive rules in Eq. 5. Since the number of recursions directly determines the sequence lengths, we determine the escape probability regarding the length of action sequences. For notational simplicity, we denote the escape probabilities $p_{i,\epsilon}^{\Omega}$ by $p$, omitting superscripts and subscripts. The expectation of the number of recursions is calculated by:

$$\lim_{n \to \infty} \sum_{k=1}^{n} kp(1-p)^k = \lim_{n \to \infty} \frac{(1-p)(1 + (1-p)^n - np(1-p)^n)}{p}, \tag{15}$$

$$= \frac{1-p}{p}. \tag{16}$$

Since we derive the escape probability when $n^{\text{rec}} > 1$, the expected number of recursions is equal to $\bar{N} - 1$:

$$\frac{1-p}{p} = \bar{N} - 1, \tag{17}$$

, where $\bar{N}$ is the average length of action sequences. Finally, we obtain the escape probability by

$$p = \frac{1}{\bar{N}}, \tag{18}$$

where this equation is used in Eq. 13 when $n^{\text{rec}} > 1$. The derivation of the escape probability $p_{\epsilon}^{\text{M}}$ of the middle variable $V^{\text{M}}$ in Eq. 6 is also formulated as the same.

# B Breadth-first Earley Parser (BEP)

## B.1 Earley parser

The Earley parser [7] is a classic algorithm that efficiently parses strings for context-free grammar. It operates by maintaining a set of states of the parsing process. Each state consists of a production rule, a position within that rule, and a position in the input string. The parser builds a parse tree for the input string, which records the structure of parsing. The Earley parser is commonly used for natural language processing tasks, such as syntactic analysis and semantic parsing.

The Earley parser consists of three main operations: scanning, prediction, and completion.

- **Scanning:** The parser matches a terminal symbol in the input string with the current position in the production rule. This operation moves the parser forward in the input string.

- **Prediction:** The parser expands a variable in the production rule based on the current position. It adds new states to the set of states for possible future matches.

- **Completion:** When the parser reaches the end of the production rule, it searches for other states predicting the head variable of the current rule. Subsequently, the parser update the positions within the rule of the searched states.

By iterating these three operations, the Earley parser builds a parse chart that represents all possible parse trees for the input string.

## B.2 Implementation details

The parsing probability $p(\boldsymbol{F}_{1:t} \to \boldsymbol{a} \,|\, G)$ (Eq. 9) can suffer from numerical underflow due to its exponential decrease as $t$ increases. To overcome the issue, we compute the probabilities in logarithmic space, following [31]. For simplicity, we denote $\log(p(\boldsymbol{F}_{1:t-1} \to \boldsymbol{a} \,|\, G))$ as $P_N$ and $\log(p(\boldsymbol{F}_{1:t-1} \to \boldsymbol{a}_{1:N-1} \,|\, G))$ as $P_{N-1}$ below:

$$P'_{N-1} = \log(g(x|\boldsymbol{a}_{1:N-1}, G)) + P_{N-1}, \tag{19}$$

$$z = \max(P_N, P'_{N-1}), \tag{20}$$

$$\log(p(\boldsymbol{F}_{1:t} \to \boldsymbol{a} \,|\, G)) = \log(\boldsymbol{Y}_{t,x}) + z + \log(\exp(P_N - z) + \exp(P'_{N-1} - z)). \tag{21}$$

$$S \rightarrow A\ B\ C$$
$$A \rightarrow A_1\ A_2$$
$$A_1 \rightarrow x_1\ [0.7] \mid x_2\ [0.3]$$
$$A_2 \rightarrow A_3\ A_4\ [0.5] \mid \epsilon\ [0.5]$$
$$A_3 \rightarrow x_4\ A_4\ [0.5] \mid \epsilon\ [0.5]$$
$$A_4 \rightarrow x_3\ A_3\ [0.5] \mid \epsilon\ [0.5]$$
$$B \rightarrow x_5\ [1.0]$$
$$C \rightarrow x_6\ [0.7] \mid x_7\ [0.3]$$

Figure 6: Toy grammar used for the example of the BEP parsing

For the computational efficiency, we set the sampling stride of input matrix $Y$ as 50 for Breakfast and 100 for 50Salads. Additionally, we set the maximum length of the refined action sequence as 20 for Breakfast and 25 for 50Salads.

### B.3 Parsing algorithm

Algorithm 1 shows the parsing procedure of BEP. We utilize a priority queue that sorts the elements in ascending order. The $currentSet$ stores multiple states with the same $m$, $n$, and $d$. See B.4 for the examples. BEP stops parsing when the probability of $a^*$ has the highest probability compared to states in the queue while ensuring the current state can reach depth 1 with only completions.

### B.4 Parsing example

In this section, we provide an example to help understand how BEP works. For simplicity, we assume that frame-wise class probabilities from the segmentation model are identical across all action classes. First of all, we define toy grammar as shown in Figure 6. In the context of grammar, $S$ indicates the starting variable. $A$, $B$, $C$, and $A_i$ for $i = [1, 2, 3, 4]$ represent the variables, while $x_j$ for $j = [1, 2, 3, 4, 5, 6, 7]$ represent terminals.

Table 9 is the history of parsing with the toy grammar. It shows the currently popped state, the visiting order, the parsed prefix, the previous state, and which states are currently in the queue. The three consecutive numbers in the column pop, from, and queue indicate $m$, $n$, and $d$ of the state. The column $p$ represents the prefix probability excluding the frame-wise probability, which can be considered a parsing probability since all frame-wise probabilities are assumed to be the same. Note that the table includes some history after the parsed sequence satisfied the early stop constraint to illustrate how BEP prioritizes the states. Returning to the subject, the table shows BEP preferentially searches for states with a small depth. In order 14, even though the probability of state $Q(1, 1, 3)$ is higher, BEP visits the state $Q(3, 0, 2)$ with a lower depth.

**Algorithm 1:** Breadth-first Earley Parser (BEP)

**Input :** probability matrix $\boldsymbol{Y}$, grammar $G$, queue size $N^{\text{queue}}$

**Output:** Best parsed sequence $\boldsymbol{a}^*$

1 **function** Breadth-first Earley Parser
2    $q \leftarrow priorityQueue()$ ;                            `// init priority queue`
3    $Q(0,0,0) \leftarrow (\Gamma \to R, Q(0,0,0), \epsilon, 1.0)$ ;             `// set initial state`
4    $q.push(0, (1.0, 0, 0, \epsilon, Q(0,0,0)))$ ;          `// push initial state to queue`
5    $\boldsymbol{a}^* \leftarrow \epsilon$ ;                                              `// init` $\boldsymbol{a}^*$
6    **while** $(d, (p(\boldsymbol{a}_{1:|\boldsymbol{a}|-1}), m, n, \boldsymbol{a}_{1:|\boldsymbol{a}|-1}, currentSet)) \leftarrow q.pop()$ **do**
7      **for** $(r, Q(i,j,k), \boldsymbol{a}, p(\boldsymbol{a}...)) \in currentSet$ **do**
         `// update` $\boldsymbol{a}^*$ `when` $\boldsymbol{a}$ `has higher probability`
8          **if** $p(\boldsymbol{a}) > p(\boldsymbol{a}^*)$ **then**
9              $\boldsymbol{a}^* \leftarrow \boldsymbol{a}$
10          **end if**
         `// prediction`
11          **if** $r$ *is* $(A \to \alpha \cdot B\beta)$ **then**
12              **for** *each* $(B \to \Gamma)$ *in* $G$ **do**
13                  $r' \leftarrow (B \to \cdot\Gamma)$
14                  $Q' \leftarrow (r', Q(m,n,d), \boldsymbol{a}, p(\boldsymbol{a}...))$
15                  $Q(m,n,d+1).add(Q')$ $q.push(d+1, (p(\boldsymbol{a}...), m, n, \boldsymbol{a}, Q(m,n,d+1)))$
16              **end for**
17          **end if**
         `// scanning`
18          **if** $r$ *is* $(A \to \alpha \cdot x\beta)$ **then**
19              $r' \leftarrow (A \to \alpha x \cdot \beta)$
20              $n' \leftarrow |Q(m+1)|$
21              $Q' \leftarrow (r', Q(i,j,k), \boldsymbol{a} + x, p((\boldsymbol{a}+x)...))$
22              $Q(m+1, n', d).add(Q')$
23              $q.push(d, (p(\boldsymbol{a}+x)..., m+1, n', d, Q(i,j,k)))$
24          **end if**
         `// completion`
25          **if** $r$ *is* $(B \to \Gamma\cdot)$ **then**
26              **for** *each* $((A \to \alpha \cdot B\beta), Q(i',j',k'), \boldsymbol{a}, p(\boldsymbol{a}...))$ *in* $Q(i,j,k)$ **do**
27                  $r' \leftarrow (A \to \alpha B \cdot \beta)$
28                  $Q' \leftarrow (r', Q(i',j',k'), \boldsymbol{a}, p(\boldsymbol{a}...))$
29                  $Q(m,n,d-1).add(Q')$
30                  $q.push(d-1, (p(\boldsymbol{a}...), m, n, \boldsymbol{a}, Q(m,n,d-1)))$
31              **end for**
32          **end if**
33      **end for**
     `// early stop when` $\boldsymbol{a}^*$ `has the highest probability and finished parsing`
34      **if** $p(\boldsymbol{a}^*) > p(\boldsymbol{a}')$ *for all* $\boldsymbol{a}'$ *in* $q$ **then**
35          **if** $\boldsymbol{a}^*$ *has parsed* **then**
36              **return** $\boldsymbol{a}^*$
37          **end if**
38      **end if**
     `// Queue pruning`
39      **if** $|q| > N^{\text{queue}}$ **then**
         `// sort` $q$ `in probability descending order`
40          $q' \leftarrow \text{sorted}(q, \text{key} = p(\boldsymbol{a}...), \text{reverse} = True)$
41          $q.clear()$
42          **for** $i \leftarrow 1$ *to* $N^{\text{queue}}$ **do**
43              $q.push(q'.pop())$
44          **end for**
45      **end if**
46    **end while**
47    **return** $\boldsymbol{a}^*$
48 **end function**

Table 9: Parsing log for the given toy grammar through BEP.

| pop | order | $m$ | $n$ | $d$ | rule | prefix | operation | from | $p$ | queue |
|---|---|---|---|---|---|---|---|---|---|---|
| - | 1 | 0 | 0 | 0 | $\Gamma \to \cdot S$ | - | ROOT | - | 1 | 000 |
| 000 | 2 | 0 | 0 | 1 | $S \to \cdot ABC$ | - | PRED | 000 | 1 | 001 |
| 001 | 3 | 0 | 0 | 2 | $A \to \cdot A_1 A_2$ | - | PRED | 001 | 1 | 002 |
| 002 | 4 | 0 | 0 | 3 | $A_1 \to \cdot x_1$ | - | PRED | 002 | 0.7 | 003 |
| | | 0 | 0 | 3 | $A_1 \to \cdot x_2$ | - | PRED | 002 | 0.3 | |
| 003 | 5 | 1 | 0 | 3 | $A_1 \to x_1 \cdot$ | $x_1$ | SCAN | 003 | 0.7 | 103, 113 |
| | 19 | 1 | 1 | 3 | $A_1 \to x_2 \cdot$ | $x_2$ | SCAN | 003 | 0.3 | |
| 103 | 6 | 1 | 0 | 2 | $A \to A_1 \cdot A_2$ | $x_1$ | COMP | 103 | 0.7 | 113, 102 |
| 102 | 7 | 1 | 0 | 3 | $A_2 \to \cdot A_3 A_4$ | $x_1$ | PRED | 102 | 0.35 | 103, 113 |
| | | 1 | 0 | 3 | $A_2 \to \cdot e$ | $x_1$ | PRED | 102 | 0.35 | |
| 103 | - | 1 | 0 | 4 | $A_3 \to \cdot a 4 A_4$ | $x_1$ | PRED | 103 | 0.175 | 203, 113, 104 |
| | | 1 | 0 | 4 | $A_3 \to \cdot e$ | $x_1$ | PRED | 103 | 0.175 | |
| | 8 | 2 | 0 | 3 | $A_2 \to e \cdot$ | $x_1$ | SCAN | 103 | 0.35 | |
| 203 | 9 | 2 | 0 | 2 | $A \to A_1 A_2 \cdot$ | $x_1$ | COMP | 203 | 0.35 | 202, 113, 104 |
| 202 | 10 | 2 | 0 | 1 | $S \to A \cdot BC$ | $x_1$ | COMP | 202 | 0.35 | 201, 113, 104 |
| 201 | 11 | 2 | 0 | 2 | $B \to \cdot x_5$ | $x_1$ | PRED | 201 | 0.35 | 202, 113, 104 |
| 202 | 12 | 3 | 0 | 2 | $B \to x_5 \cdot$ | $x_1\ x_5$ | SCAN | 202 | 0.35 | 302, 113, 104 |
| 302 | 13 | 3 | 0 | 1 | $S \to AB \cdot C$ | $x_1\ x_5$ | COMP | 302 | 0.35 | 301, 113, 104 |
| 301 | 14 | 3 | 0 | 2 | $C \to \cdot x_6$ | $x_1\ x_5$ | PRED | 301 | 0.245 | 302, 113, 104 |
| | | 3 | 0 | 2 | $C \to \cdot x_7$ | $x_1\ x_5$ | PRED | 301 | 0.105 | |
| 302 | 15 | 4 | 0 | 2 | $C \to x_6 \cdot$ | $x_1\ x_5\ x_6$ | SCAN | 302 | 0.245 | 402, 412, 113, 104 |
| | 17 | 4 | 1 | 2 | $C \to x_7 \cdot$ | $x_1\ x_5\ x_7$ | SCAN | 302 | 0.105 | |
| 402 | 16 | 4 | 0 | 1 | $S \to ABC \cdot$ | $x_1\ x_5\ x_6$ | COMP | 402 | 0.245 | 401, 412, 113, 104 |
| 401 | - | - | - | - | $\Gamma \to S \cdot$ | $x_1\ x_5\ x_6$ | COMP | 401 | 0.245 | 412, 113, 104 |
| 412 | 18 | 4 | 1 | 1 | $S \to ABC \cdot$ | $x_1\ x_5\ x_7$ | COMP | 412 | 0.105 | 411, 113, 104 |
| 411 | - | - | - | - | $\Gamma \to S \cdot$ | $x_1\ x_5\ x_7$ | COMP | 411 | 0.105 | 113, 104 |
| 113 | - | 1 | 1 | 2 | $A \to A_1 \cdot A_2$ | $x_2$ | COMP | 113 | 0.3 | 112, 104 |

# C   Comparison with the existing grammar induction algorithms

## C.1   Existing grammar induction algorithms

Kuehne *et al.* [23] introduce a hierarchical context-free grammar induction algorithm. The root rule with the starting variable $S$ is induced as $S \to V_1 \,|\, V_2 \,|\, ... \,|\, V_{N^{\mathrm{A}}}$, where the variable $V_i$ represents a single activity and $N^{\mathrm{A}}$ is the number of activities from the dataset. Then each $V_i$ expands into action sequences from each activity, *i.e.*, the rule is formed as: $V_i \to \mathcal{A}_{i,1} \,|\, \mathcal{A}_{i,2} \,|\, ... \,|\, \mathcal{A}_{i,|\mathcal{A}_i|}$, where $\mathcal{A}_i$ is a set of action sequences from the $i$-th activity and each $\mathcal{A}_{i,j}$ represents a $j$-th action sequence in $\mathcal{A}_i$.

Richard *et al.* [35] propose a grammar induction method for a probabilistic right-regular grammar, where every rule has the form of $\tilde{H} \to c\, H$. The algorithm is motivated by n-gram models [18] and finite grammars [14]. Specifically, the variable $\tilde{H}$ represents an action sequence $\boldsymbol{a}_{1:n}$, $H$ represents an action sequence $\boldsymbol{a}_{1:n-1}$, and a terminal $c$ is an action class of $a_n$. The induced grammar can express the intermediate action sequences $\boldsymbol{a}_{1:n}$ and expands its rules based on the sequential order of actions.

Recently, Qi *et al.* [32] adopt the Automatic Distillation of Structure (ADIOS) [37] algorithm to induce a probabilistic context-free grammar. The ADIOS algorithm finds the significant patterns (AND rules) and equivalence action classes (OR rules) from the given action sequences. The algorithm identifies repetitive patterns in action sequences to minimize redundant sequences and find potential candidates for generalized action classes. Following the grammar induction methods of ADIOS, we

Table 10: The performance comparison with other grammar induction algorithms on two benchmark datasets.

| dataset | reprod. | refinement | grammar induction | edit | F1@10 | F1@25 | F1@50 | acc. |
|---------|---------|------------|-------------------|------|-------|-------|-------|------|
| 50Salads [39] | ✓ | - | - | 76.5 | 83.8 | 81.7 | 74.8 | **86.1** |
| | ✓ | ✓ | Kuehne *et al.* [23] | 62.9 | 73.0 | 70.6 | 63.0 | 78.6 |
| | ✓ | ✓ | Richard *et al.*[35] | 63.1 | 73.1 | 70.5 | 62.9 | 78.7 |
| | ✓ | ✓ | ADIOS-AND | 58.3 | 70.0 | 68.0 | 59.4 | 76.2 |
| | ✓ | ✓ | ADIOS-OR | 61.1 | 72.0 | 70.1 | 62.4 | 78.9 |
| | ✓ | ✓ | KARI | **79.9** | **85.4** | **83.8** | **77.4** | 85.3 |
| Breakfast [23] | ✓ | - | - | 75.6 | 77.3 | 72.0 | 59.4 | **74.3** |
| | ✓ | ✓ | Kuehne *et al.* [23] | 72.8 | 74.0 | 69.1 | 55.5 | 72.9 |
| | ✓ | ✓ | Richard *et al.*[35] | 77.3 | 77.2 | 72.2 | 59.4 | 74.1 |
| | ✓ | ✓ | ADIOS-AND | 69.2 | 69.8 | 64.9 | 52.2 | 72.4 |
| | ✓ | ✓ | ADIOS-OR | 70.3 | 71.8 | 66.8 | 54.2 | 71.8 |
| | ✓ | ✓ | KARI | **77.8** | **78.8** | **73.7** | **60.8** | 74.0 |

set a decreasing ratio of the motif extraction algorithm $\eta$ to 1, a significance level for the decrease ratio $\gamma$ to 0.1, and the context window size 1 for ADIOS-AND-induced grammar. For ADIOS-OR-induced grammar, we set $\eta$ to 0.9, $\gamma$ to 0.1, and the context window size to 4.

However, none of these approaches have managed to effectively integrate recursive rules, which are crucial for representing intricate and lifelike structures of action phrases and activities. The proposed KARI algorithm introduces a probabilistic context-free grammar that allows for the expression of complex activity structures, which captures a distinctive temporal structure based on key actions.

### C.2 Performance on temporal action segmentation

In Table 10, we compare the performance of each grammar induction algorithm on refining temporal action segmentation models [45]. The overall results show that the KARI-induced grammar demonstrates the best refinement performance compared to the other grammar induction algorithms, showing a significant performance gap in both datasets. The induced grammar of Richard *et al.*also shows better performance than other grammar induction algorithms except for KARI, indicating that the ability to represent intermediate action sequences by production rules helps improve refinement performance. In conclusion, the generalization capabilities and variability of expressing action sequences are essential to guide the temporal action segmentation network to better refinement results.

## D   Qualitative results

We provide additional qualitative results for the Breakfast and 50Salads. Figure 7 shows examples of successful output refinements by the KARI-induced grammar, demonstrating its ability to cover various sequences comprising combinations of multiple actions. This is further evident in Figure 7a, where ADIOS-OR falls short in covering the *'add dressing'* action following the *'serve salad'* action, while KARI handles it proficiently.

We also show failure cases of the KARI-induced grammar in Figure 8, where further improvement is needed. We acknowledge that the KARI-induced grammar sometimes deletes certain actions. This deletion of actions, along with the challenges posed by inaccurate identification of actions by the segmentation model, show areas for improvement in the refinement process. We recognize these as opportunities for future work to enhance the performance of the grammar induction algorithm and address these limitations.

## E   Broader Impact

The research presented in this paper holds significant potential for impact across multiple domains. The development of efficient and effective grammar induction algorithms for activity grammar, coupled with the Breadth-first Earley parser, has the potential to greatly enhance human activity recognition and understanding systems. This, in turn, can have far-reaching implications in various

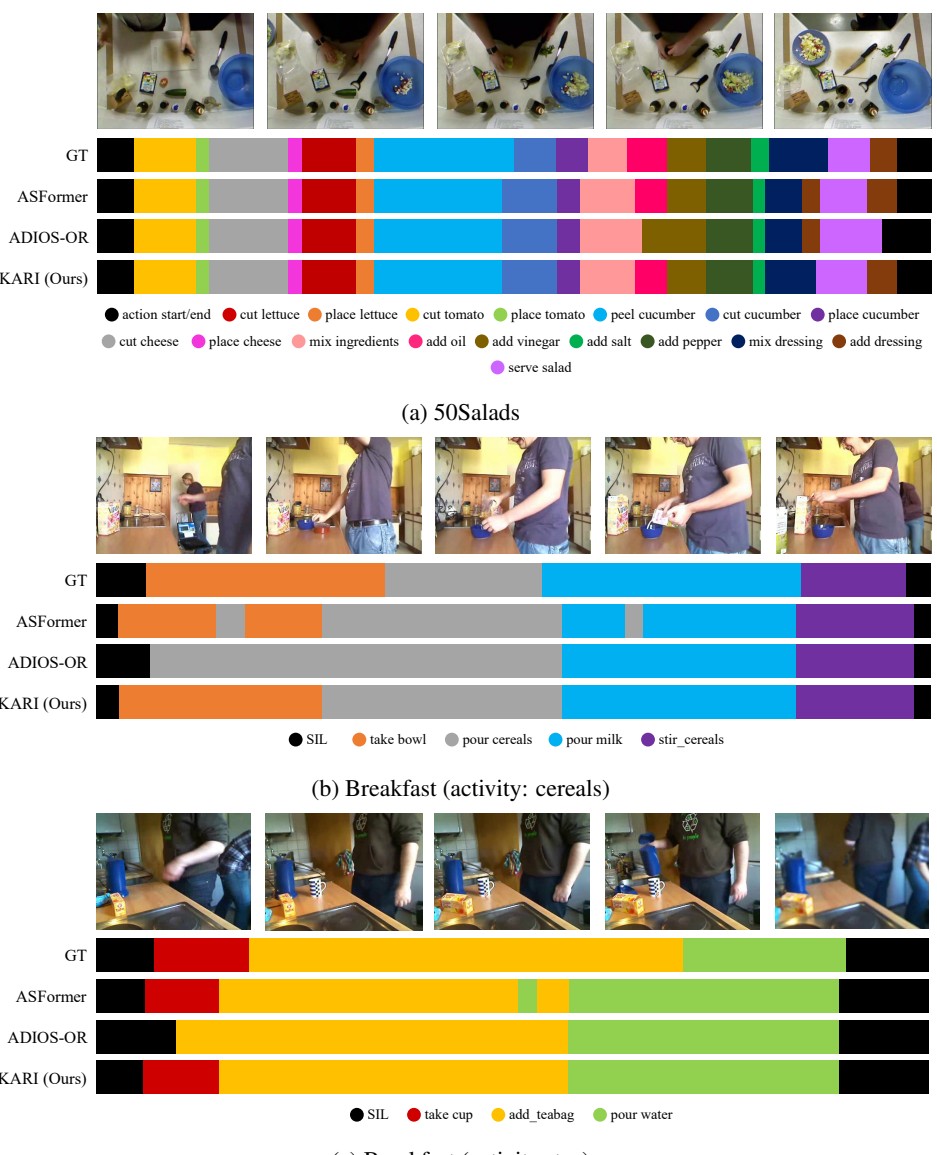

(a) 50Salads

(b) Breakfast (activity: cereals)

(c) Breakfast (activity: tea)

Figure 7: Qualitative results on successful cases

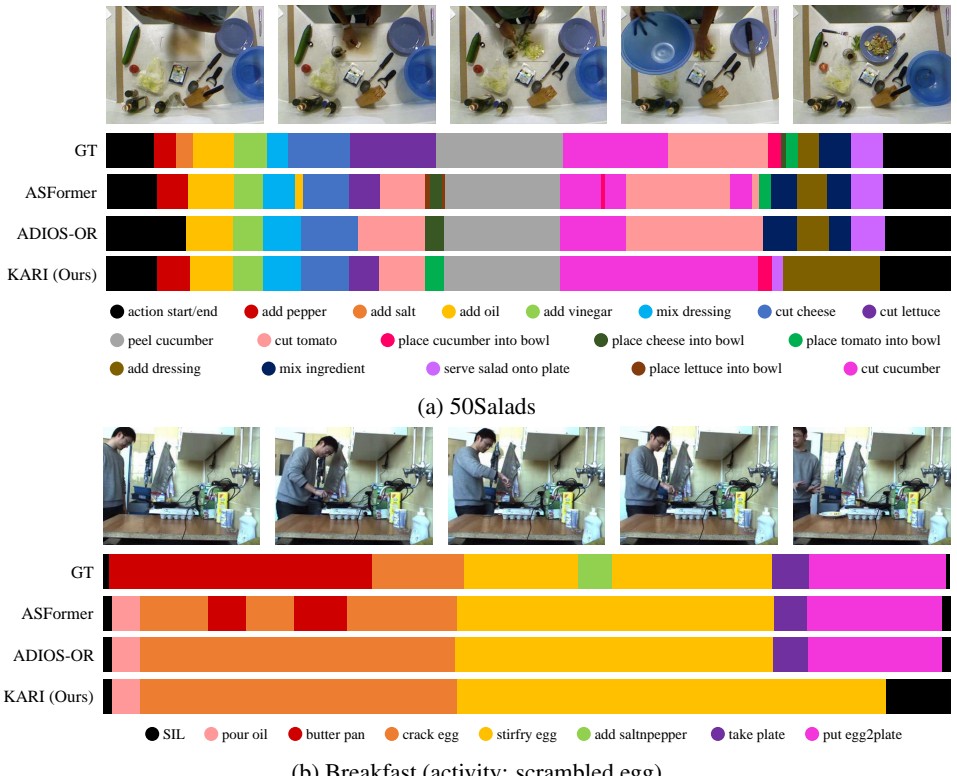

(a) 50Salads

(b) Breakfast (activity: scrambled egg)

Figure 8: Qualitative results on failure cases

applications, such as video surveillance, human-computer interaction, robotics, and healthcare monitoring. By improving the accuracy and efficiency of activity recognition systems, our research contributes to advancements in these domains, enabling more robust and intelligent systems. The broader implications of this research extend beyond activity grammar induction itself, fostering innovation and enhancing the capabilities of intelligent systems in diverse fields.

