# OpenReview forum: "Activity Grammars for Temporal Action Segmentation"
_NeurIPS.cc/2023/Conference — NeurIPS 2023 poster_

### Official Review · Reviewer_rBwB · 2023-07-07

**Soundness:** 2 fair
**Presentation:** 2 fair
**Contribution:** 2 fair
**Rating:** 5
**Confidence:** 5

**Summary:**

This paper proposes a grammar-based activity segmentation method. The authors proposed a grammar induction algorithm, as well as an improvement over the existing activity grammar parser, the Generalized Earley Parser (GEP). The proposed model shows improvements over prior works on both grammar induction and action segmentation.

**Strengths:**

- The exploration of marrying neural-symbolic representations is an essential aspect of research, especially given the limited application of neural-symbolic methods for real-world problems.

- The proposed grammar induction method does improve existing prior works and could potentially be beneficial for future research.

**Weaknesses:**

- The experiments on activity segmentation are mainly compared with baselines that are not state-of-the-art. Though the ablations prove the improvement in grammar induction and parsing, the overall performance can not be justified by the current experiments (i.e., might need better results to justify the motivation of grammar-based activity understanding methods).

- Another concern is the design of grammar. Given recent advances in language modeling and unsupervised grammar induction, the motivation of symbolic methods is not very clear or not shown by the experiments.

- The overall formulation and algorithm design for the BEP follows from the GEP parser.

- The notation used in this paper is error-prone and needs to be clearer, especially given that the grammar-based method largely depends on these notations. (e.g. L.146, should it be $a_i^{M}$).

**Questions:**

See weakness section

**Limitations:**

The authors have properly addressed the limitations.

---

> ### Author Rebuttal · Authors · 2023-08-10
>
> We thank reviewer rBwB for their meaningful comments.
>
> ### **[Comparison with SOTA]**
> Following the reviewer's suggestion, we compare ours with the state-of-the-art methods in Table R3 where ours shows comparable or superior performance. Note that most of the refinement methods [A3, 9, A5, 5, A6, 15, 41] typically fine-tune their underlying temporal action segmentation models, whereas our method does not require such fine-tuning, thus being able to apply to any black-box temporal action segmentation models. The breakfast and 50salads benchmarks are the two most standard benchmarks for action segmentation. We will add more experiments to the final manuscript using another benchmark.
>
> ### **[Motivation of our approach]**
>
> Similar to other neuro-symbolic approaches [A7], our grammar-based method facilitates an explainable output using its grammar structure, improving the generalization performance to unseen action sequences, which we have demonstrated in Tables 2-3, Figure 4, Table S1, and Figure S3.
>
> Furthermore, combining a symbolic approach with neural networks can also increase robustness and reliability [A8]. In our case, the induced grammar is used to improve the noisy output of a temporal action segmentation network. To better see the effect, we conduct experiments in Table R2 using the ASFormer that performs multiple refinement stages in decoding [8, 42]; as shown in the first compartment of Table R2, the performance of the lower decoding stages shows lower performance.
> The second compartment of Table R2
> shows the results of applying our method to each stage of the decoder in ASFormer.
> It reveals that our grammar-based refinement brings more significant improvement when initial action segmentation is from lower stages, i.e., less accurate.
>
> ### **[Effectiveness of BEP over GEP]**
> While both BEP and GEP are based on the Earley parser, they exhibit differences in their search algorithms.
> In detail, BEP and GEP have distinct orders for exploring production rules, leading to differences in the searching space as we applied pruning techniques to BEP. For better understanding, we explain how the depth-based priority works during the parsing through the example in Table S1 and Section B.4 of the supplementary material.
>
> In addition, the motivation of BEP stems from the fact that GEP was unable to parse the CFG generated through KARI within a reasonable time (lines 175-177).
> To address the challenge, we have introduced additional pruning techniques and confirmed that the searching algorithm proposed in BEP was more effective (Table 4), prompting us to adopt it as the preferred solution.
>
> ### **[Clarification on KARI]**
> Thanks for the correction. $\boldsymbol{a}^\mathrm{M_i}$ should be revised to $\boldsymbol{a}_i^\mathrm{M}$ in line 146. We will try our best to clarify notations and explanations in the final version of our paper.

---

> > ### Author Response · Authors · 2023-08-21
> >
> > ### **[ Additional comparison with SOTA - Applying our method to DTL [41] ]**
> > In addition to Table R3 of the pdf attached to the global response, we apply our method to DTL [41] on 50 Salads. Since its pre-trained weights are not yet released despite our request, we attempted to reproduce DTL incorporated with MSTCN [8] and ASFormer [42] based on the original paper and its official code repository [A9].
> >
> > The reproduced results of DTL are summarized in Table R4. For both Table R4-(a) and R4-(b), the first, second, and third rows report performance of the baseline (either MSTCN or ASFormer), that of the combination of DTL and the baseline, and that of applying ours to the combination.
> > The tables demonstrate that our method substantially improved performance of DTL in all the segmentation metrics (i.e., edit score and F1 scores). This suggests that **our grammar refinement method is effective for DTL, regardless of the baseline incorporated with it**; it indeed offers complementary benefits to various temporal action segmentation models, such as MSTCN, ASFormer, and DTL.
> >
> > **[Table R4. The performance of applying our method to DTL]**
> >
> > (a) MSTCN
> >
> > | model                        | edit | F1@10 | F1@25 | F1@50 | acc |
> > |------------------------------|------|-------|-------|-------|-----|
> > |MSTCN (reprod.) |62.4|69.5|65.3|55.7|75.2|
> > |MSTCN + DTL [41] (reprod.) |67.3|74.9|72.7|64.7|79.9|
> > |MSTCN + DTL [41] + ours |68.4 (1.1&uarr;)|76.7 (1.8&uarr;)|74.8 (2.1&uarr;)|65.5 (0.8&uarr;)|78.9 (1.0&darr;)|
> >
> > (b) ASFormer
> >
> > | model                        | edit | F1@10 | F1@25 | F1@50 | acc |
> > |------------------------------|------|-------|-------|-------|-----|
> > |ASFormer (reprod.) |76.5|83.8|81.7|74.8|86.1|
> > |ASFormer + DTL [41] (reprod.) |78.8|85.1|84.2|76.3|86.3|
> > |ASFormer + DTL [41] + ours |80.2 (1.4&uarr;) |85.9 (0.8&uarr;)|84.8 (0.6&uarr;)|77.6 (1.3&uarr;)|85.4 (0.9&darr;)|
> >
> >
> > [A9] Ziwei Xu et al. DTL-action-segmentation. https://github.com/ZiweiXU/DTL-action-segmentation, 2022.

---

> > > ### Comment · Reviewer_rBwB · 2023-08-22
> > > **Post-rebuttal response**
> > >
> > > The authors have addressed most of my concerns, therefore I'm willing to increase my original rating to 5.

---

> > > > ### Author Response · Authors · 2023-08-22
> > > >
> > > > We again thank reviewer rBwB for the motivating feedback and are glad to hear that most of the concerns are addressed by our rebuttal. We will include results from the rebuttal in the final manuscript.

---

### Official Review · Reviewer_FXdk · 2023-07-07

**Soundness:** 2 fair
**Presentation:** 3 good
**Contribution:** 2 fair
**Rating:** 5
**Confidence:** 3

**Summary:**

This paper proposes a new grammar induction algorithm, an effective parser, and a grammar evaluation framework. They improve temporal action segmentation by extracting and handling context-free grammar with recursive rules. The assessment presents good generalization and discrimination capabilities of induced grammar.

After reading the rebuttal, my concern about the second and third points are resolved. For the first point, the authors provide the SOTA table as requested. However, the performance of the proposed method is not convincing.

Overall, given that this paper introduces a new framework for temporal action segmentation, I still consider the pros outweigh the cons and will keep my initial recommendation.

**Strengths:**

1. KARI proposes a new grammar induction algorithm, an effective parser, and a grammar evaluation framework. They improve temporal action segmentation by extracting and handling context-free grammar with recursive rules. The assessment presents good generalization and discrimination capabilities of induced grammar.

2. KARI outperforms state-of-the-art models on two benchmarks, Breakfast, and 50 Salads, for the temporal action segmentation task.

3. The writing is good.

**Weaknesses:**

1. The evaluation of MIF is based on breakfast and 50salads. Can the author provide results on more benchmarks? The SOTA table is missing in the paper.
2. The method is relatively new, but the details are mostly presented in the form of text. Some illustrations for the method's visual representation are missing. I think this is very beneficial to readers' understanding and the dissemination of the paper.
3. For the first row of MS-TCN results in Table 2, is better than KARI, but the author did not bold it. Also, the author should give an explanation why KARI doesn't work here.

**Questions:**

See weaknesses.

In the first row of Section 3.1, the citation is missing.

**Limitations:**

Limitations are not discussed in the paper.

---

> ### Author Rebuttal · Authors · 2023-08-10
>
> We thank reviewer FXdk for constructive comments.
>
> ### **[Comparison with SOTA]**
> Following the reviewer's suggestion, we compare ours with the state-of-the-art methods in Table R3 where ours shows comparable or superior performance. Note that most of the refinement methods [A3, 9, A5, 5, A6, 15, 41] typically fine-tune their underlying temporal action segmentation models, whereas our method does not require such fine-tuning, thus being able to apply to any black-box temporal action segmentation models. The breakfast and 50salads benchmarks are the two most standard benchmarks for action segmentation. We will add more experiments to the final manuscript using another benchmark.
>
> ### **[Grammar illustration]**
> Following the suggestion, we created Figure R1 with example sequences and KARI-induced grammar for a better understanding. We also provide the induction example in Section A.3 of the supplementary material.
> In Figure R1-(a), we give an example sequence of $\boldsymbol{a}$ of the 'milk' activity, where 'spoon powder' and 'pour milk' are the key actions, and visualize how the sequences are divided into sub-strings according to the key actions.
> Figure R1-(b) presents the KARI-induced grammar from $\boldsymbol{a}$.
>
> ### **[MS-TCN results in Table 2]**
> Let us clarify that KARI works on MS-TCN in Table 2.
> The first row of each compartment indicates the performance of the original paper.
> We apply the proposed methods to the reproduced models, as the pre-trained model of MS-TCN is not open-sourced.
> Therefore, we compare the performance of the proposed methods with the reproduced performance in the second row of each compartment, similar to the previous work [1, 14, 41].

---

> > ### Author Response · Authors · 2023-08-21
> >
> > ### **[ Additional comparison with SOTA - Applying our method to DTL [41] ]**
> > In addition to Table R3 of the pdf attached to the global response, we apply our method to DTL [41] on 50 Salads. Since its pre-trained weights are not yet released despite our request, we attempted to reproduce DTL incorporated with MSTCN [8] and ASFormer [42] based on the original paper and its official code repository [A9].
> >
> > The reproduced results of DTL are summarized in Table R4. For both Table R4-(a) and R4-(b), the first, second, and third rows report performance of the baseline (either MSTCN or ASFormer), that of the combination of DTL and the baseline, and that of applying ours to the combination.
> > The tables demonstrate that our method substantially improved performance of DTL in all the segmentation metrics (i.e., edit score and F1 scores). This suggests that **our grammar refinement method is effective for DTL, regardless of the baseline incorporated with it**; it indeed offers complementary benefits to various temporal action segmentation models, such as MSTCN, ASFormer, and DTL.
> >
> > **[Table R4. The performance of applying our method to DTL]**
> >
> > (a) MSTCN
> >
> > | model                        | edit | F1@10 | F1@25 | F1@50 | acc |
> > |------------------------------|------|-------|-------|-------|-----|
> > |MSTCN (reprod.) |62.4|69.5|65.3|55.7|75.2|
> > |MSTCN + DTL [41] (reprod.) |67.3|74.9|72.7|64.7|79.9|
> > |MSTCN + DTL [41] + ours |68.4 (1.1&uarr;)|76.7 (1.8&uarr;)|74.8 (2.1&uarr;)|65.5 (0.8&uarr;)|78.9 (1.0&darr;)|
> >
> > (b) ASFormer
> >
> > | model                        | edit | F1@10 | F1@25 | F1@50 | acc |
> > |------------------------------|------|-------|-------|-------|-----|
> > |ASFormer (reprod.) |76.5|83.8|81.7|74.8|86.1|
> > |ASFormer + DTL [41] (reprod.) |78.8|85.1|84.2|76.3|86.3|
> > |ASFormer + DTL [41] + ours |80.2 (1.4&uarr;) |85.9 (0.8&uarr;)|84.8 (0.6&uarr;)|77.6 (1.3&uarr;)|85.4 (0.9&darr;)|
> >
> >
> > [A9] Ziwei Xu et al. DTL-action-segmentation. https://github.com/ZiweiXU/DTL-action-segmentation, 2022.

---

### Official Review · Reviewer_CeJ9 · 2023-07-07

**Soundness:** 3 good
**Presentation:** 3 good
**Contribution:** 3 good
**Rating:** 5
**Confidence:** 3

**Summary:**

This paper addresses the challenge of temporal action segmentation by introducing an activity grammar to guide neural predictions. The proposed approach involves a grammar induction algorithm (KARI) to extract a powerful context-free grammar from action sequence data. Additionally, an efficient generalized parser (BEP) transforms frame-level probabilities into a reliable sequence of actions based on the induced grammar. Experimental results on benchmark datasets show significant improvements in both performance and interpretability of temporal action segmentation.

**Strengths:**

This paper is well written which demonstrates its motivation, methodology and experiments. Especially, the method of this paper is easy to follow and the provided visualization is a plus to understand the proposed algorithm.

The idea of introducing the recursive rules is inspiring, which helps to identify the repetitions of actions and action phrases.

The proposed grammar evaluation scheme demonstrates the effectiveness of the proposed KARI-induced activity grammar, achieving good recall while maintaining reasonable precision.

The experimental results look promising, achieving good performance on various benchmarks.


**Weaknesses:**

The benchmark is relatively small, which generates concerns about the scalability of the proposed method. Also, it’s good to know the computational cost of the proposed method to evaluate its scalability.

The design of the proposed KARI is not well justified by using ablation study. It will be good to show the effectiveness of each component and its alternatives.

Minor:

Missing reference for ln 109 and 119.


**Questions:**

Is the proposed method limited by the action type? For example, it could only work for well-structured activities.

**Limitations:**

as is mentioned in previous sections.

---

> ### Author Rebuttal · Authors · 2023-08-10
>
> We thank reviewer CeJ9 for their meaningful comments and suggestions.
>
> ### **[Scalability and computation cost of the proposed method]**
>
> Following the reviewer's suggestion, we analyze the computational cost of the KARI and BEP.
> Since KARI induces activity grammars based on the activity sequences in the dataset, the time consumption is directly affected by the number of sequences. The Breakfast dataset includes 196 unique action sequences from 1,712 video sequences.
>
> Figure R3-(a) shows the running time plot when gradually increasing the dataset from 40\% to 100\%, where we can empirically estimate linear running time with respect to the dataset size; it takes less than 0.1 seconds to induce grammar from the entire activity sequences in our implementation.
> Figure R3-(b) shows a similar plot when varying the number of sequences from 200 to 1,000.
> These results indicate that KARI has reasonable scalability so that it can be applied to large-scale datasets.
>
> The computational cost of BEP depends on the complexity of the grammar $G$ generated from the induction algorithm.
> Regarding BEP, the time complexity of computing the parsing probability for each sequence is $O(T)$ [29] where $T$ denotes the number of frames, resulting in a worst-case time complexity for the entire parsing process becomes $O(T|G|)$.
>
> ### **[Ablation study on KARI]**
>
> Please refer to the general response for the ablation study on KARI.
>
> ### **[Limited to well-structured activities?]**
>
> KARI induces an activity grammar by analyzing action sequences using flexible temporal dependencies across actions; it uses OR rules for temporally equivalent actions and AND rules for dependent actions. It does not require specific activity structures in action sequences, showing impressive generalization to real-world data (Table S2) as well as random synthetic data (Table 1).
>
> ### **[Missing reference]**
>
> Thanks for letting us know. We will add citations of [A1] in line 109 and [A2] in line 119.

---

### Official Review · Reviewer_YXrU · 2023-07-09

**Soundness:** 3 good
**Presentation:** 2 fair
**Contribution:** 3 good
**Rating:** 5
**Confidence:** 3

**Summary:**

This paper presents a grammar induction algorithm that takes as input sequences of frame-level predictions and outputs structured sequences of actions. The advantage of their approach is that it allows recursive rules, which enhances its generalization abilities.

**Strengths:**

1. The method is more flexible than previously proposed grammar induction methods.
2. The results are better compared to baselines.
3. All the steps are explained in detail and most of them are well justified.

**Weaknesses:**

1. The grammar explanation (section 3.2) is confusing. Especially the large amount of superindices and subindices representing different concepts. For example, what does E^{M(m,n)} mean? Or, in "t \in {b, a, k(m, n)}" what does the "b" represent? Probably simpler notation or a figure would help. I also did not understand what "each sub-string a_i^M starts with a key action and includes all the key actions in K" means. Do you start at a key action and do not close the sub-string until all key actions have been found? But then, I assume there will only be one such sub-string in the whole sequence. Is there any overlap between sub-string? Is it possible that a sequence does not contain any key action?

2. I do not think the synthetic activity grammars are a good setting to judge the quality of the proposed approach. Without delving into the details of the generated random grammars, it is unclear whether the biases introduced in the generation may make them easier to deal with for some kinds of methods than others. Real-world grammars do not have that problem.

3. There are no results showing the importance of the specific contribution. Did the authors identify cases where the inclusion of recursive rules made a difference?

Minor weakness: a few typos and grammar mistakes: "due to the reason" -> "due to this reason" (lines 4, 22), "and also be applied" -> "and can also be applied" (line 29), "we proceeds" (line 137), "has" -> "have" (line 70), "other researches" (line 74), a few empty citations (lines 109, 119), etc.

**Questions:**

- Would it be possible to combine this approach to perform end-to-end training with the frame-level predictions Y? This is, train the grammar and the temporal action segmentation network simultaneously.

- Accuracy does not improve when using grammars (it actually decreases). Why is that the case? I would assume that having contextual grammatical information about future and past actions would improve the accuracy of the current action prediction. Do you think the accuracy would improve with a flexible/good enough grammar, or this is a limitation of using action grammars?

**Limitations:**

Limitations and broader impact are discussed in the appendix.

---

> ### Author Rebuttal · Authors · 2023-08-10
>
> We thank reviewer YXrU for constructive comments and suggestions.
>
> We will clearly revise Section 3.2 using simpler notations and more illustrations.
>
> ### **[Grammar illustration]**
> Following the suggestion, we created Figure R1 with example sequences and KARI-induced grammar for a better understanding. We also provide the induction example in Section A.3 of the supplementary material.
> In Figure R1-(a), we give an example sequence of $\boldsymbol{a}$ of the 'milk' activity, where 'spoon powder' and 'pour milk' are the key actions, and visualize how the sequences are divided into sub-strings according to the key actions.
> Figure R1-(b) presents the KARI-induced grammar from $\boldsymbol{a}$.
>
> ### **[Partitioning strategy for sub-string $\boldsymbol{a}^\mathrm{M}_i$]**
> In lines 143-144, the statement, "each sub-string $\boldsymbol{a}_i^\mathrm{M}$ starts with a key action and includes all the key actions in $\mathcal{K}$," indicates that the sub-string $\boldsymbol{a}_i^\mathrm{M}$ should include all of the key actions existed in the activity.
> In the example in Figure R1-(a), the middle sequence $\boldsymbol{a}^\mathrm{M}$ is further divided into two distinct sub-strings, denoted as $\boldsymbol{a}^\mathrm{M}_1$ and $\boldsymbol{a}^\mathrm{M}_2$;
> $\boldsymbol{a}^\mathrm{M}_1$ consists of ['spoon powder', 'stir milk', 'pour milk', 'stir milk'] and $\boldsymbol{a}^\mathrm{M}_2$ comprises ['pour milk', 'spoon powder', 'pour milk'].
> Each sub-string starts with a key action, extends until all key actions of the activity have been included, and ends before the key action of the next sub-strings begins. As can be seen in this example, multiple sub-strings, i.e., $\boldsymbol{a}^\mathrm{M}_1$ and $\boldsymbol{a}^\mathrm{M}_2$, that contain the entire key actions, can exist in a single sequence $\boldsymbol{a}$.
>
> ### **[Middle variable $E^{\mathrm{M}(m,n)}$]**
> In line 156, $E^{\mathrm{M}(m,n)}$ indicates a variable for the rule that covers an action sub-string between the $n_\mathrm{th}$ key action and the $n+1_\mathrm{th}$ key action when the $m_\mathrm{th}$ permutation $\pi_m$ is applied to key actions.
> Since the temporal order between key actions can vary, we introduce the concept of key action permutations $\Pi^\mathrm{M}$ (line 146).
> In the example of Figure R1 with two key actions of 'pour milk' and 'spoon powder', there exist two possible permutations: $\pi_1=$ ['pour milk', 'spoon powder'] and $\pi_2=$ ['spoon powder', pour milk'], where $\Pi^\mathrm{M}=\{\pi_1, \pi_2\}$.
> The rule of $E^\mathrm{M(1,1)}$ thus has 'stir milk' as a body part.
> This action occurs between 'spoon powder' and 'pour milk,' i.e., the first action of $\pi_1$ and the second action of $\pi_1$.
>
> ### **[Action sequences without key actions]**
> It is possible to extract grammar from the sequences without any key actions by simply putting the entire action sequence $\boldsymbol{a}$ into $\boldsymbol{a}^\mathrm{L}$.
>
> ### **[Evaluation on synthetic \& real dataset]**
> We generated synthetic activity grammars based on the properties of real-world action sequences, which can be observed from existing activity datasets [20, 36]. The procedure for grammar generation can be summarized as follows. We randomly select one to five key actions among 20 terminal elements and randomly determine the number of variables. For production rules, we assign each terminal to variables in a random manner (lines 275-279) and generate rules by randomly compositing them with 'OR' and 'AND'.
>
> In addition, we also conducted experiments using grammars without any key actions; 50 synthetic grammars are generated and evaluated with the same procedure in the main paper. As shown in the table below, the KARI-induced grammar shows powerful generalization performance compared to ADIOS-induced grammar, while precision is comparable with others. We also visualize the confusion matrix in Figure R2 for better understanding.
>
> |grammar|precision|recall|
> |---|---|---|
> |ADIOS-AND|1.00|0.09|
> |ADIOS-OR|1.00|0.60|
> |KARI (ours)|0.84|1.00|
>
> While the analyses with diverse synthetic grammars demonstrate the efficacy of KARI, we agree with the reviewer's concern that synthetic activity grammar may introduce some biases that are different from real-world activity grammars. Note that we thus also did evaluate the grammar induction algorithms on the real-world dataset, as shown in Table S2 of the supplementary material.
>
> ### **[Ablation study on KARI]**
> Please refer to the general response for the ablation study on KARI.
>
> ### **[End-to-end learning with KARI]**
> In this paper, the proposed method is applied to the pre-trained models without fine-tuning.
> Since we do not incorporate grammar into the models in training, end-to-end training is currently unavailable.
> However, we believe that a learnable version of KARI would make it possible to be trained with the neural network regarding the previous work [25, 26], and we leave it as future work.
>
> ### **[Accuracy drop]**
> To analyze the slight decrease in accuracy while significantly increasing edit and F1 scores, we conduct a case study.
> We present the quantitative and qualitative results of a single sample in Figure R4 and the table below:
> |model|acc|edit|F1@10|F1@25|F1@50|
> |---|---|---|---|---|---|
> |ASFormer[42]|91.87|85.71|85.0|85.0|77.5|
> |Ours|90.86(1.01$\downarrow$)|100.0(14.29$\uparrow$)|95.78(10.28$\uparrow$)|95.78(10.28$\uparrow$)|87.32(9.82$\uparrow$)|
>
> From the red box drawn in Figure R4, we observe that this discrepancy seems to be caused by adjusting the boundary between actions based on the segmentation output.
> Future improvements could be achieved by adding boundary regression modules [15] or other comparable methods.
>
> ### **[Typos and empty citations]**
> In line 160, $t\in\{\mathrm{b}, \mathrm{a}, \mathrm{}(m,n)\}$ should be revised to $t\in\{\mathrm{L}, \mathrm{R}, \mathrm{M}(m,n)\}$ which indicates a set of superscripts of notations.
>
> We will add citations of [A1] in line 109 and [A2] in line 119.

---

### Author Rebuttal · Authors · 2023-08-10

We thank all the reviewers for their insightful comments and suggestions.
We are happy to see that the reviewers have given our work a positive evaluation, noting that "the method is more flexible than previously proposed grammar induction methods (YXrU)," "the idea of introducing the recursive rules is inspiring (CeJ9)," "the assessment presents good generalization and discrimination capabilities(FXdk)," and "the proposed grammar induction method does improve existing prior works and could potentially be beneficial for future research(rBwB)."

Nevertheless, the reviewers also point out important comments that:

1. the explanation of grammar induction can be improved for clarity with illustrations,
2. the grammar induction algorithm requires ablation study,
3. comparison with the state of the arts needs to be added,
4. motivation for the proposed method needs to be clarified.

Through this rebuttal, we aim to clearly expound the components of KARI and their respective roles, compare the proposed method with the state-of-the-art models, and explain the motivation of our approach.
We will revise the manuscript by incorporating the detailed comments from the reviewers.
We include Figures R1-R4 and Tables R1-R3 for additional results in the pdf files.

In response to the questions posed by reviewers YXrU and CeJ9 regarding the ablation study of KARI, the results are included in the general response.

### **[Ablation study on KARI]**
We conduct an ablation study to evaluate the two main components of KARI, key action extraction and recursive rule generation. The results are shown in Table R1, which demonstrates the effectiveness of both components.
In particular, eliminating recursive rules (third row in Table R1) significantly decreases the overall performance. We will add these results to our final manuscript.


### **[References]**

[A1] J. Frederick et al. Basic methods of probabilistic context free grammars. Springer Berlin Heidelberg 1992.

[A2] D. Klein and CD. Manning. A generative constituent-context model for improved grammar induction. ACL 2002.

[A3] D. Wang et al. Temporal relational modeling with self-supervision for action segmentation. AAAI Conference on Artificial Intelligence. 2021.

[A4] S.J. Li et al. MS-TCN++: Multi-stage temporal convolutional network for action segmentation. TPAMI 2020.

[A5] Z. Wang et al. Boundary-aware cascade networks for temporal action segmentation. ECCV 2020.

[A6] D. Singhania et al. Coarse to fine multi-resolution temporal convolutional network. arXiv. 2021.

[A7] P. Hitzler et al. Neuro-symbolic approaches in artificial intelligence, National Science Review. Volume 9, Issue 6. 2022.

[A8] R. Evans and E. Grefenstette. Learning explanatory rules from noisy data. Journal of Artificial Intelligence Research 61 (2018): 1-64.

---

> ### Author Response · Authors · 2023-08-18
> **A gentle reminder**
>
> Dear reviewers,
>
> We'd like to thank again for your effort and time dedicated to our submission. We've addressed your concerns in our rebuttal, and it would be very helpful if you could give us any further thoughts and update your scores before the author-reviewer discussion phase ends. Your opinion would be invaluable to us in improving our work, and we would be glad to respond further to your questions. Thank you for your consideration.
>
> Best regards,
> Authors

---

### Decision · Program_Chairs · 2023-09-21

**Decision:**

Accept (poster)

**Comment:**

This paper received four unanimous borderline accept recommendations. The reviewers generally appreciated the direction of the paper and acknowledged the empirical performance improvement over the baselines. However, there were several concerns about the adequacy of the experiments, such as the absence of ablation studies demonstrating the importance of the specific contributions (recursive rules and symbolic method), the focus on synthetic data instead of real data, the lack of evaluation on larger datasets, and the absence of a comparison with state-of-the-art approaches. The rebuttal provided additional results that addressed most of these concerns. The only remaining concern pertains to the absence of results on large-scale benchmarks, as raised by `FXdk`. This meta-reviewer recognizes this concern but also understands that conducting large-scale experiments may be challenging without access to substantial computational resources.

Overall, this meta-reviewer finds the main idea presented in the paper to be inspiring and experiments to be convincing to justify the claims made in the paper. The only remaining concern is the small-scale nature of the experiments, which puts the scalability of the approach in question. However, the rebuttal provided evidence showing the scalability (as shown in response to `CeJ9`). Given this, we recommend acceptance.